# *GoodDiffusion*: Proactive Copyright Protection for Diffusion Bridge Models via Learnable Sample-specific Signatures

**Shixi Qin** [1]   **Zhiyong Yang\*** [1]   **Shilong Bao** [1]   **Zitai Wang** [2]   **Qianqian Xu** [2 3]   **Qingming Huang\*** [1 2]

## Abstract

This paper tackles the challenging problem of developing a proactive copyright protection mechanism that cuts off unauthorized use of diffusion bridge models. Existing studies largely fall into post-hoc attribution (e.g., watermarking and fingerprinting) or degradation-only defenses, which offer only **indirect and limited** preventive effect. We therefore propose *GoodDiffusion*, inspired by backdoor mechanisms, to enforce **model-level** use-time control by internalizing authorization into the generative process through a *selectively permissive, otherwise closed* behavior. Specifically, *GoodDiffusion* preserves high-quality generation for authorized queries carrying valid signatures, yet **refuses to generate** for unauthorized inputs. We further theoretically show that naive static-signature designs (like conventional backdoor injection) are fundamentally fragile, since a surrogate signature can be efficiently recovered via gradient-based optimization. To strengthen security, we introduce a *Learnable Signature Network* (LSN) that assigns *sample-specific* signatures conditioned on each input. This breaks the universality of signatures and prevents a surrogate from transferring across inputs. Extensive experiments validate that *GoodDiffusion* effectively blocks unauthorized use while maintaining strong generation quality for authorized users. The code is available at https://github.com/qsx830/GoodDiffusion.

[1]School of Computer Science and Technology, University of Chinese Academy of Sciences, Beijing 101408, China [2]State Key Laboratory of AI Safety, Institute of Computing Technology, Chinese Academy of Sciences, Beijing, China [3]Beijing Academy of Artificial Intelligence (BAAI), Beijing, China. Correspondence to: Zhiyong Yang <yangzhiyong21@ucas.ac.cn>, Qingming Huang <qmhuang@ucas.ac.cn>.

*Proceedings of the 43rd International Conference on Machine Learning*, Seoul, South Korea. PMLR 306, 2026. Copyright 2026 by the author(s).

## 1. Introduction

Diffusion-based generative models are now widely used to produce high-quality images in products and creative workflows (Ho et al., 2020; Song & Ermon, 2020; Song et al., 2020; Nichol & Dhariwal, 2021; Dhariwal & Nichol, 2021; Karras et al., 2022), making *copyright protection (CP)*[1] (Chen et al., 2022a) a practical requirement for real-world deployment. Without safeguards, these models can mass-produce unlicensed derivatives, harming creators' livelihoods and causing direct financial loss. This has spurred growing demand (Zhang et al., 2018) to deter and prevent unauthorized use.

Inspired by earlier CP successes in deep learning, most diffusion-model defenses currently rely on **output-level watermarking or fingerprinting** (Zhao et al., 2023; Xie et al., 2025; Wang et al., 2025b; Gai et al., 2025), i.e., embedding imperceptible yet detectable signals in generated images for later attribution. While valuable for traceability, such watermark-based protection is **passive**: it provides only **after-the-fact evidence rather than real-time prevention**, allowing unauthorized generations to spread at scale before detection and enforcement can catch up. A meaningful step toward mitigating this issue is the recently proposed PCDiff (Gai et al., 2025), which introduces a key-conditioned module that proactively *degrades* output quality under unauthorized use. However, we argue that such a degradation-only strategy may be **insufficient** in the wild: it does **not fully cut off unauthorized generation**, where thieves can still obtain usable images (albeit at reduced quality). Therefore, we ask the following question:

> *Can we develop a proactive copyright protection mechanism that **completely prevents** unauthorized generation at the **model-level**?*

In this paper, we present *GoodDiffusion*, a first step toward answering this question. Inspired by the adage *"No Ticket, No Ride,"* we aim to endow diffusion models with proactively model-level control over authorized (with tickets)

---

[1]The law of the U.S. has a comprehensive introduction for copyright protection: https://www.ce9.uscourts.gov/jury-instructions/node/257.

versus unauthorized (without tickets) generation requests. In other words, *GoodDiffusion* seeks to learn a *selectively permissive, otherwise closed* condition (C): it preserves high-quality generation under authorized use, yet **refuses to generate** under unauthorized or out-of-control queries. This shifts CP from post-hoc output level to **model-level prevention** internalized in the generative process itself. More interestingly, we find this principle to be conceptually aligned with backdoor mechanisms (Chou et al., 2023; Zhai et al., 2023; Qin et al., 2025), where a diffusion model can be steered into producing incorrect outputs when a malicious trigger is embedded in the input.

Motivated by this connection, we first introduce a naive *backdoors-for-good* baseline that uses a fixed *perturbation* pattern (rather than an explicit trigger) to pursue the above *conditional behavior-switching* mode (i.e., (C)). Taking **diffusion bridge models** (Liu et al., 2023; Zhou et al., 2024) for image-to-image (I2I) tasks as an example, we train the model to produce high-quality outputs only when the input carries a predefined signature (implemented as a fixed perturbation), and to emit a predefined warning response for clean, unlicensed queries. Despite its apparent effectiveness, we theoretically show that this naive design is fragile in practice since a malicious adversary with full access to the model can efficiently recover the signature via gradient-based optimization (Theorem 4.1).

To address this issue, we further develop a *Learnable Signature Network (LSN)* to leverage dynamic, sample-specific signatures conditioned on each input. The difference between the fixed signature and the sample-specific signatures can be viewed from a geometric perspective. It is known that images lie on a low-dimensional manifold embedded in the high-dimensional pixel space (Carlsson, 2009). Intuitively, a fixed signature can be interpreted as a fixed vector in the pixel space, leading to **a parallel shift of the entire image manifold**. On the contrary, the sample-specific signatures produce input-dependent perturbations, resulting in **a nonlinearly deformed image manifold** that cannot be represented by a simple vector addition. This input-dependent design also echoes the broader observation in long-tailed learning that real-world data often contains sparse or tail regions where uniform operations may be suboptimal (Wang et al., 2024a;c; Zhao et al., 2024; 2025; 2026). Empirical analysis validates the intuition that there exists no universal surrogate signature that can be recovered in this sample-specific design. Even if a thief obtains a signature for a specific input, the signature cannot be transferred to other inputs, thus enhancing the security of our method.

We conduct extensive experiments of three representative I2I tasks (i.e., super-resolution, inpainting, and deblurring) on CelebA and ImageNet datasets with various diffusion bridge models to validate the effectiveness of *GoodDiffu-*

*sion*. *GoodDiffusion* achieves strong protection behavior for unauthorized usages, while preserving satisfactory generation quality under authorized inferences.

Overall, our contributions can be summarized as follows:

- **An Early Trial for Proactive CP.** We propose *GoodDiffusion*, a model-level prevention framework that internalizes authorization into the generative process to proactively cut off unauthorized generation.

- **Some Theoretical Insights.** We show that *static* signature designs are fundamentally fragile, as one can efficiently recover a surrogate signature via gradient-based optimization (Theorem 4.1). In contrast, our learnable *sample-specific* signatures mitigate this vulnerability by preventing a transferable surrogate signature.

- **Extensive Evaluation.** Comprehensive experiments validate that *GoodDiffusion* achieves strong protection performance while preserving high-quality generation for authorized users.

## 2. Related Work

**Diffusion Bridge Models.** Diffusion models (Ho et al., 2020; Song & Ermon, 2020; Song et al., 2020; Nichol & Dhariwal, 2021; Dhariwal & Nichol, 2021; Karras et al., 2022) have achieved remarkable success in image generation and various downstream tasks (Chung et al., 2023; Huang et al., 2024; Yu et al., 2024; Cheng et al., 2025). Building on this progress, diffusion bridge models (DBM) (Liu et al., 2023; Wang et al., 2025a) generalize the paradigm by building stochastic processes between two arbitrary distributions. In particular, diffusion Schrödinger bridge models (De Bortoli et al., 2021; Chen et al., 2022b; Shi et al., 2023; Deng et al., 2024b; Qiu et al., 2025) realize entropy-regularized optimal transport. Apart from these, DDBM (Zhou et al., 2024) and its variants (Zheng et al., 2025; He et al., 2024) build diffusion bridges via Doob's $h$-transform (Doob & Doob, 1984), and parameterize the drift term with a score-based model trained by denoising score-matching (Vincent, 2011).

**Passive Protection.** Watermarking techniques have been widely adopted for copyright protection (Liu et al., 2022; Chen et al., 2023). Prior works on diffusion models (Xie et al., 2025) offer *passive* defenses by embedding specific patterns as watermarks into the Gaussian noise input (Wen et al., 2023; Yang et al., 2024b), the diffusion process (Yang et al., 2024a), or the generated images (Zhao et al., 2023; Peng et al., 2025). Other watermarking methods integrate discriminative watermarks into the diffusion process (Fernandez et al., 2023; Feng et al., 2024; Min et al., 2024; Wang et al., 2025b). Once the predefined watermarks are detected, the model owner can claim the copyright. Beyond water-

marking, (Deng et al., 2024a) proposes a non-transferable learning mechanism, which enables the diffusion model to be resistant to fine-tuning.

**Proactive Protection.** In the context of traditional discriminative models, researchers explored encrypting model parameters or architectures (Lin et al., 2020; Xue et al., 2022; Zhou et al., 2023; Mu et al., 2024; Sun et al., 2025) to *proactively* prevent unauthorized access. In addition, some works adopt backdoor attacks for CP (Li et al., 2024; 2025), where the model produces correct predictions only when a specific trigger is present in the inputs. Such proactive protection has been extended to multimodel datasets (Zhang et al., 2025). As for diffusion models, some works apply cryptographic techniques (Chen & Yan, 2024; Yao, 2024; He et al., 2025; Guo et al., 2025) to proactively ensure the privacy and security of data, prevent the abuse of diffusion models in downstream applications (Liu et al., 2025), but they do not directly protect the copyright of diffusion models. Recently, PCDiff (Gai et al., 2025) proposes an encryption module in diffusion models to degrade the unauthorized generation quality. Despite its proactive quality degradation, PCDiff does not fully deny unauthorized generation.

# 3. Preliminary

In this paper, we study the copyright protection (CP) in a representative Image-to-Image (I2I) **diffusion bridge model** (DBM) (Zhou et al., 2024), which directly takes images as inputs and generates target images. The key idea is to implement a special **backdoor attack** that injects sample-specific signatures into the inputs. Although we take DBM as an example, the core idea of proactive CP can be easily extended to other types of diffusion models (e.g., text-to-image models) by designing proper backdoor attacks, which is discussed in Appendix B.9.

In this section, we first introduce the preliminaries of DBM and backdoor attacks. Then, we formalize the overall settings of the CP problem.

## 3.1. Diffusion Bridge Models

The DBM model aims to construct a diffusion process between two arbitrary distributions, enabling direct image generation conditioned on a source image distribution. Obviously, the DBM model is suitable for multiple image I2I tasks, such as super-resolution, inpainting, and deblurring.

Formally, given a pair of images sampled from a joint distribution $(\boldsymbol{x}_0, \boldsymbol{x}_1) \sim p(\boldsymbol{x}_0, \boldsymbol{x}_1)$, the DBM aims to build the forward and backward diffusion processes via the Doob's $h$-transform (Doob & Doob, 1984; Rogers & Williams, 2000). One can parameterize the diffusion bridge model via the

denoising score-matching (DSM) objective (Vincent, 2011):

$$\mathcal{L}(\boldsymbol{\theta}) = \mathbb{E}_{t,\boldsymbol{x}_0,\boldsymbol{x}_1,\boldsymbol{x}_t} \left[ \lambda(t) \left\| s_{\boldsymbol{\theta}}(\boldsymbol{x}_t, t) - s^*(\boldsymbol{x}_t, t) \right\|_2^2 \right], \quad (1)$$

where $s^*(\boldsymbol{x}_t, t) = \nabla_{\boldsymbol{x}_t} \log p(\boldsymbol{x}_t)$ is the ground-truth score function, $s_{\boldsymbol{\theta}}(\boldsymbol{x}_t, t)$ is the learnable model with parameters $\boldsymbol{\theta}$, and $\lambda(t)$ is a time-dependent weighting function. After training, one can sample from the learned bridge using standard reverse-time SDE or probability-flow ODE solvers (Ho et al., 2020; Song et al., 2021; Lu et al., 2022).

## 3.2. Backdoor Attacks on Diffusion Models

Backdoor attacks (Chou et al., 2023; Zhai et al., 2023; Qin et al., 2025) aim to manipulate the behavior of diffusion models at inference. Formally, the attacker injects specific trigger patterns $\boldsymbol{k}$ into the training data $\boldsymbol{x}_1$ as poisoned inputs: $\tilde{\boldsymbol{x}}_1 = \boldsymbol{x}_1 + \boldsymbol{k}$. Accordingly, given a clean training set $D = \{(\boldsymbol{x}_1, \boldsymbol{x}_0)^i\}_{i=1}^N$, the poisoned training set $\tilde{D} = \{(\tilde{\boldsymbol{x}}_1, \bar{\boldsymbol{x}}_0)^i\}_{i=1}^N$ can be constructed, where $\bar{\boldsymbol{x}}_0$ is a target image specified by the attacker. Then, the diffusion model is jointly trained on $D$ and $\tilde{D}$ to learn the backdoor behavior as follows:

$$\mathcal{L}_{\text{total}}(\boldsymbol{\theta}) = (1 - \pi) \cdot \mathcal{L}_D(\boldsymbol{\theta}) + \pi \cdot \mathcal{L}_{\tilde{D}}(\boldsymbol{\theta}), \quad (2)$$

where $\pi \in (0, 1)$ is the poison rate, and $\mathcal{L}_D(\boldsymbol{\theta})$ and $\mathcal{L}_{\tilde{D}}(\boldsymbol{\theta})$ are the denoising score-matching losses in Eq. 1 calculated on the datasets $D$ and $\tilde{D}$ respectively. At inference time, the backdoored diffusion model produces the predefined output $\bar{\boldsymbol{x}}_0$ with poisoned inputs $\tilde{\boldsymbol{x}}_1$; otherwise, it yields normal outputs $\boldsymbol{x}_0$ for clean inputs $\boldsymbol{x}_1$.

## 3.3. Threat Model and Protection Goals

In this paper, we propose to protect the copyright of a DBM against unauthorized model thieves. We summarize the model thief's capabilities and the protection goals as follows.

**Model Thief's Capabilities.** We assume a strong white-box model thief who has full access to the model parameters and architecture from the authorized user. However, since the signature service is separately maintained by the model owner, the thief cannot access the legitimate signature service. In addition, we assume the thief has limited computational resources (e.g., it is not feasible to brute-force all possible signatures), otherwise the thief does not need to steal the model but train a new one from scratch.

**Protection Goals.** Overall, the protection goals can be summarized as follows:

(G1) The diffusion model can generate **high-quality** images exclusively when the valid signatures are provided.

(G2) The protection method should be **secure** against malicious model thieves **in the white-box scenario**.

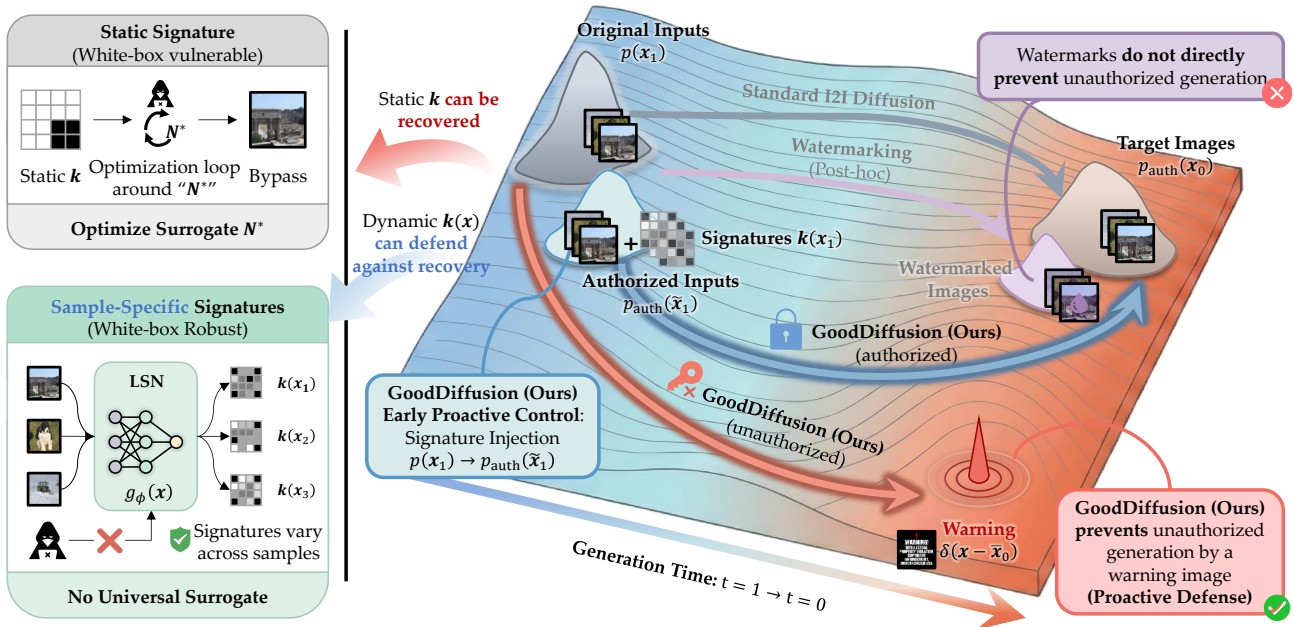

*Figure 1.* **Overview of *GoodDiffusion*.** (*Left*) The static signature design is vulnerable in white-box scenarios, while the sample-specific signature design can effectively defend against malicious model thieves. (*Right*) The trained DBM integrates two diffusion bridges. The authorized trajectory (blue) follows $p_{\mathrm{auth}}(\tilde{\boldsymbol{x}}_1) \to p_{\mathrm{auth}}(\boldsymbol{x}_0 \mid \tilde{\boldsymbol{x}}_1)$. The unauthorized trajectory (red) follows $p_{\mathrm{warn}}(\boldsymbol{x}_1) \to p_{\mathrm{warn}}(\bar{\boldsymbol{x}}_0)$. Although watermarks can be used for passive protection, the gap between watermarked and non-watermarked images may be imperceptible to human eyes, thus still allowing unauthorized usage. However, *GoodDiffusion* provides early proactive control: the unauthorized user can only generate the warning image, which is completely different from the target images.

## 4. *GoodDiffusion*

### 4.1. Motivation

We seek a proactive copyright protection (CP) method for diffusion bridge models (DBM). Compared to watermark defenses, proactive protection aims to directly prevent unauthorized usage at the very beginning of the diffusion process.

To implement this, the key challenge is to keep the protection mechanism effective in the **white-box** scenario, where the model thief has already obtained an executable copy of the model, including the model weights, architecture, and additional items for model encryption. In such an extreme case, any defense relying on parameter encryption is brittle.

To overcome this challenge, we manage to provide a **model-level** protection from the perspective of *generative process* rather than *model parameters*. Our key insight is that the diffusion trajectory can be manipulated via backdoor attacks on diffusion models (Sec. 3.2). Specifically, one can steer the diffusion trajectory converging to different endpoints conditioned on whether the input contains backdoor triggers. Inspired by this, we propose to **bind** the diffusion trajectory with our copyright protection mechanism. As the protection mechanism is *implicitly* embedded in the diffusion trajectory, even if the *explicit* model parameters and architecture are leaked, the model thief cannot control the

diffusion trajectory for unauthorized uses.

Given this motivation, how to design such a backdoor attack for CP? Next, we formalize this idea into the proposed *GoodDiffusion*, starting from an intuitive, naive implementation, and then enhancing its security in white-box scenarios.

### 4.2. Backdoor for Good

Existing backdoor attacks in diffusion models aim to mislead the model to produce incorrect outputs. In contrast, the *GoodDiffusion* proposes to leverage backdoor attacks for *good* purposes, i.e., to protect the copyright of DBM. In particular, the core idea is to **reverse** the generation objectives of conventional backdoor attacks: the DBM generates high-quality outputs with triggered inputs; otherwise, it yields incorrect images. As such, we prepare a special kind of perturbations as a signature $\boldsymbol{k}$, and a predefined warning image indicating unauthorized usages, similar to the conventional backdoor attacks (Sec. 3.2).

Accordingly, given clean paired images $\{(\boldsymbol{x}_1, \boldsymbol{x}_0)^i\}_{i=1}^N$, we build an **authorized training set** $\tilde{D} = \{(\tilde{\boldsymbol{x}}_1, \boldsymbol{x}_0)^i\}_{i=1}^N$, and a **warning set** $\bar{D} = \{(\boldsymbol{x}_1, \bar{\boldsymbol{x}}_0)^i\}_{i=1}^{N'}$, where $\tilde{\boldsymbol{x}}_1 = \boldsymbol{x}_1 + \boldsymbol{k}$ denotes the authorized input, and $\bar{\boldsymbol{x}}_0$ is the predefined warning image. The DBM is jointly trained on $\tilde{D}$ and $\bar{D}$ to learn the backdoor behavior. At each training iteration, we draw the branch $B \in \{\mathrm{auth}, \mathrm{warn}\}$ with $\Pr(B = \mathrm{auth}) =$

$\pi_k$ and $\Pr(B = \text{warn}) = 1 - \pi_k$. Similar to Eq. 2, we derive the following training objective for *GoodDiffusion*:

$$\mathcal{L}_{\text{total}}(\boldsymbol{\theta}) = \pi_k \cdot \mathcal{L}_{\tilde{D}}(\boldsymbol{\theta}) + (1 - \pi_k) \cdot \mathcal{L}_{\bar{D}}(\boldsymbol{\theta}), \qquad (3)$$

where $\mathcal{L}_{\tilde{D}}(\boldsymbol{\theta})$ and $\mathcal{L}_{\bar{D}}(\boldsymbol{\theta})$ are the denoising score-matching losses in Eq. 1 calculated on $\tilde{D}$ and $\bar{D}$, respectively.

The trained DBM integrates two distinct diffusion bridges into a single network: One authorized bridge mapping the authorized input distribution $p_{\text{auth}}(\tilde{\boldsymbol{x}}_1)$ to high-quality target images $p_{\text{auth}}(\boldsymbol{x}_0 \mid \tilde{\boldsymbol{x}}_1)$, and one unauthorized bridge mapping clean inputs $p_{\text{warn}}(\boldsymbol{x}_1)$ to the warning image $p_{\text{warn}}(\bar{\boldsymbol{x}}_0)$, as illustrated in Fig. 1. Thus, the trained DBM exhibits the desired backdoor behavior: it generates high-quality target images $\tilde{\boldsymbol{x}}_0$ if and only if authorized inputs $\tilde{\boldsymbol{x}}_1$ are provided, thus achieving the protection goal (G1).

### 4.3. A Naive Implementation

As a naive implementation, we implement the signature $\boldsymbol{k}$ as a subtle, fixed Gaussian perturbation. Let $M \in \{0, 1\}^{H \times W}$ be a binary mask that indicates a small region for the signature injection, and let $\sigma_e > 0$ be a minor noise level. The signature injection can be expressed as:

$$\tilde{\boldsymbol{x}}_1 = \boldsymbol{x}_1 + \boldsymbol{k} = \boldsymbol{x}_1 + M \odot \boldsymbol{\eta}, \quad \boldsymbol{\eta} \sim \mathcal{N}(0, \sigma_e^2 \boldsymbol{I}). \quad (4)$$

This design minimally distorts $\boldsymbol{x}_1$ yet reliably injects the signature. Till now, we have presented the *GoodDiffusion* for CP. While this design appears to meet the protection objectives (G1), a question remains: *Is it secure against an unauthorized adversary in the white-box scenario?*

### 4.4. Security of the Naive Implementation

In the regime of backdoor attacks, trigger inversion techniques (Tao et al., 2022; Jiang et al., 2025), aiming to detect and remove backdoors from compromised models, have been widely studied. While prior works focus on defensive purposes, we argue that similar techniques can be exploited by the model thief to *recover* a surrogate signature to bypass our protection. In particular, we assume the white-box model thief has full access to the model architecture and parameters $s_{\boldsymbol{\theta}}$, and holds a **recovery set** $D_a = \{(\boldsymbol{x}_1, \boldsymbol{x}_0)^i\}_{i=1}^{N_a}$ containing a batch of clean samples. In addition, the thief knows the principle of signature injection is an additive perturbation, but does not possess the valid signature. Similar to prior trigger inversion works, one can formulate the signature inversion process as an optimization problem:

$$\boldsymbol{N}^* = \arg\min_{\boldsymbol{N}} \mathbb{E}\left[\lambda(t) \left\| s_{\boldsymbol{\theta}}(\hat{\boldsymbol{x}}_t, t) - s^*(\hat{\boldsymbol{x}}_t, t) \right\|_2^2\right],$$
$$\hat{\boldsymbol{x}}_1 = \boldsymbol{x}_1 + \boldsymbol{N}, \quad \hat{\boldsymbol{x}}_t \sim q(\boldsymbol{x}_t \mid \hat{\boldsymbol{x}}_1, \boldsymbol{x}_0), \qquad (5)$$

where $\boldsymbol{N} \in \mathbb{R}^{H \times W}$ is the surrogate signature to approximate the original signature $\boldsymbol{k}$. We have the following the-

orem showing that the recovered surrogate signature can effectively replace the original one.

**Theorem 4.1** (White-Box Signature Recovery). *To bypass the protection of the* GoodDiffusion *with a static signature $\boldsymbol{k}$, one can treat a whole-image perturbation $\boldsymbol{N} \in \mathbb{R}^{H \times W}$ as the surrogate attack variable. After the optimization, the score function with the surrogate signature $s_{\boldsymbol{\theta}}(\hat{\boldsymbol{x}}_t, t)$ perfectly matches the score function $s_{\boldsymbol{\theta}}(\tilde{\boldsymbol{x}}_t, t)$ with the true signature, thus the recovered surrogate signature $\boldsymbol{N}^*$ approximates the true signature $\boldsymbol{k}$.*

Once $\boldsymbol{N}^*$ is obtained, the model thief can easily generate high-quality target images by injecting the surrogate signature $\boldsymbol{N}^*$ into arbitrary inputs. Extensive experiments in Sec. 5.4 validate this risk. We will present our solution to address this risk in the next section.

### 4.5. Sample-Specific Signatures

We argue that the risk stems from the *universality* of the static signature: a single **input-independent $\boldsymbol{k}$** works for all inputs. Once the model thief finds a surrogate signature $\boldsymbol{N}$, the model thief can apply it for arbitrary inputs to bypass our protection. In other words, to mitigate the risk, the core challenge is to break this universality.

We thus propose *Sample-Specific Signatures*, where each input is injected with a unique, **input-dependent** signature. Specifically, we introduce a *Learnable Signature Network* (LSN) $g_{\boldsymbol{\phi}}(\cdot)$ that maps a plain input $\boldsymbol{x}_1$ to an input-dependent signature: $\boldsymbol{k}(\boldsymbol{x}_1) = g_{\boldsymbol{\phi}}(\boldsymbol{x}_1)$. In addition, we blend the generated signature with the original input as follows:

$$\tilde{\boldsymbol{x}}_1 = \gamma \cdot \boldsymbol{x}_1 + (1 - \gamma) \cdot \boldsymbol{k}(\boldsymbol{x}_1), \qquad (6)$$

where $\gamma \in (0, 1)$ is a hyperparameter that controls the strength of the signature injection. In this way, the authorized input $\tilde{\boldsymbol{x}}_1$ preserves more semantic information from the original input $\boldsymbol{x}_1$, thus maintaining satisfactory generation performance for authorized users.

We jointly optimize $g_{\boldsymbol{\phi}}$ and $s_{\boldsymbol{\theta}}$ under the mixture objective in Eq. 3, with the authorized branch using $\tilde{D}$ and the unauthorized branch using $\bar{D}$. Our formulation follows the broader practice of using tractable objectives to induce desired target behaviors, whose reliability has been extensively studied in the learning theory literature (Mao et al., 2023). Thus, the overall objective becomes:

$$\mathcal{L}_{\text{total}}(\boldsymbol{\theta}, \boldsymbol{\phi}) = \pi_k \cdot \mathcal{L}_{\tilde{D}}(\boldsymbol{\theta}, \boldsymbol{\phi}) + (1 - \pi_k) \cdot \mathcal{L}_{\bar{D}}(\boldsymbol{\theta}). \quad (7)$$

As $\boldsymbol{k}(\boldsymbol{x}_1)$ varies with the input, the recovery objective in Eq. 5 no longer admits a universal $\boldsymbol{N}$ that minimizes the loss for all $(\boldsymbol{x}_1, \boldsymbol{x}_0) \in D_a$. Even if the model thief optimizes a signature $\boldsymbol{N}'$ against the subset, it fails to generalize to other inputs. The sample-specific signature design effectively

*Table 1.* **CelebA: Protection Effectiveness vs Generation Quality.** We report two tasks: **Super-Resolution** and **Deblurring**. For each bridge model, we report two adjacent rows: unprotected model (Unprotected) and *GoodDiffusion* protected model. The protection effectiveness is measured by **Abuse Rate** (AR), which indicates the fraction of unauthorized inputs incorrectly mapped to high-quality target images. As the unprotected model does not perform any protection, the AR is always 100%. The generation quality is evaluated by **FID**, **PSNR**, and **SSIM**. We also report **Error Rate** (ER), which measures the fraction of authorized generation incorrectly blocked to warning images. The inpainting results are presented in Appendix B.3.

| Model | Setting | Super-Resolution | | | | | Deblurring | | | | |
|---|---|---|---|---|---|---|---|---|---|---|---|
| | | AR↓ | FID↓ | PSNR↑ | SSIM (E-02)↑ | ER↓ | AR↓ | FID↓ | PSNR↑ | SSIM (E-02)↑ | ER↓ |
| DDBM-VP | Unprotected | 100 | 12.68 | 32.43 | 89.43 | – | 100 | 5.43 | 43.99 | 98.45 | – |
| | *GoodDiffusion* | 0 | 16.22 | 32.13 | 90.31 | 0.06 | 0 | 9.49 | 36.63 | 95.71 | 0.25 |
| DDBM-VE | Unprotected | 100 | 13.42 | 32.02 | 88.87 | – | 100 | 11.14 | 39.95 | 97.31 | – |
| | *GoodDiffusion* | 0 | 28.64 | 28.03 | 84.81 | 0 | 0 | 22.01 | 33.77 | 91.71 | 0 |
| I2SB | Unprotected | 100 | 22.05 | 32.48 | 89.93 | – | 100 | 28.78 | 32.06 | 88.73 | – |
| | *GoodDiffusion* | 0 | 28.25 | 30.72 | 86.18 | 0.25 | 0 | 29.94 | 28.87 | 84.44 | 0 |
| DBIM | Unprotected | 100 | 7.88 | 32.95 | 91.25 | – | 100 | 0.88 | 45.40 | 99.54 | – |
| | *GoodDiffusion* | 0 | 13.91 | 32.67 | 91.21 | 0.13 | 0.06 | 6.34 | 38.49 | 97.08 | 0 |

addresses the risk of surrogate signature recovery in white-box scenarios, thus achieving the protection goal (G2).

### 4.6. Discussion

Here is a practical scenario of applying *GoodDiffusion* for CP. Consider a model owner who shares the model with another authorized user. At inference time, the authorized user requests valid, sample-specific signatures from the model owner for signature injection and then runs the diffusion model to generate high-quality outputs conditioned on the authorized inputs. However, a model thief may infiltrate the server of the authorized user and steal **an executable copy of the diffusion model** via model extraction attacks (Hua et al., 2018; Sun et al., 2021) or espionage, aiming to use the model without paying licensing fees. In this case, conventional access control (e.g., username/password), engineering solutions (e.g., model partitioning), or parameter encryptions are ineffective once the thief obtains the model. In contrast, the proposed *GoodDiffusion* can prevent such model theft, **as the model thief does not have legitimate signatures maintained by the model owner**. This scenario highlights *GoodDiffusion* as a practical method for white-box copyright protection of diffusion models.

We consider this scenario to be reasonable as it is aligned with the **"separation of duties"** principle in security (Groll et al., 2025). That is, the signature service is maintained separately by the model owner, while the model is deployed on the authorized user's server. Even if an attacker can access the model, they cannot access the signature service without also compromising the owner's infrastructure. This

separation is a common security practice to mitigate potential threats. In practice, modern commercial Key Management Service (KMS) implements such separation by design[2], where the clients obtain authorization from a centralized license server rather than carrying the signature service locally (Barker et al., 2007).

In real applications, most business models are equipped with powerful gateway authentication and access control mechanisms for protection in **software-level**, but once the model is leaked, the thief can use it without any restriction. *GoodDiffusion* can be a complementary solution to prevent such model theft at the **model-level**: even if the thief has an executable copy of the model, they cannot use it without valid signatures. *GoodDiffusion* can be easily integrated with existing software-level protections to provide a comprehensive defense against model theft.

## 5. Experiments

### 5.1. Experimental Setup

We conduct experiments on two datasets, CelebA (Liu et al., 2015) and ImageNet (Deng et al., 2009). On each dataset, we evaluate three representative I2I tasks: super-resolution (SR), inpainting, and deblurring.

The experiments are performed on $256 \times 256$ images for both datasets. For the super-resolution task, the low-resolution inputs ($64 \times 64$) are obtained by pooling the high-resolution

---
[2]Google Cloud KMS: `https://docs.cloud.google.com/kms/docs/separation-of-duties`

images ($256 \times 256$) with a scale factor of 4. For the inpainting task, each image is randomly masked with the $20\% - 30\%$ freeform masks (Saharia et al., 2022). For the deblurring task, the blurry inputs are generated by convolving the original images with a Gaussian kernel. The warning image $\bar{x}_0$ is set as a predefined image with warning text. In addition, we set $\pi_k = 0.5$ in Eq. 7 to balance the authorized and warning branches, and set the strength of signature injection to $\gamma = 0.9$ for authorized users.

We evaluate *GoodDiffusion* with three representative diffusion bridge models: DDBM (Zhou et al., 2024), I2SB (Liu et al., 2023), and DBIM (Zheng et al., 2025). For the DDBM, we consider two variants with different transition kernels: DDBM-VP and DDBM-VE (Song et al., 2020). We implement *GoodDiffusion* with a UNet (Ronneberger et al., 2015), the same architectures as in (Liu et al., 2023) for fair comparisons. The Learnable Signature Network $g_\phi$ for generating sample-specific signatures adopts a UNet++ model (Zhou et al., 2018).

## 5.2. Evaluation Metrics

We evaluate the performance of *GoodDiffusion* from two perspectives: protection effectiveness against unauthorized model thieves and generation quality for authorized users.

**Protection Effectiveness.** We simulate **the unauthorized behaviors by model thieves** by feeding clean inputs (i.e., without valid signatures) to the protected model, and evaluate if the model refuses to generate high-quality outputs. We report the **Abuse Rate**, defined as the fraction of the failed protections over all unauthorized generations. For each generation, the protection is considered failed if the MSE between the generated image and the target ground-truth image is below a certain threshold. Therefore, the Abuse Rate can be calculated as: AR = (Number of unauthorized inputs generating high-quality outputs) / (Total number of unauthorized inputs).

**Generation Quality.** This simulates **the authorized usages by legitimate users**, and evaluates the generation quality when valid signatures are provided. We measure the **FID** (Heusel et al., 2017), **PSNR**, and **SSIM** (Wang et al., 2004) for authorized generations. We also report the **Error Rate**, defined as the fraction of authorized inputs incorrectly mapped to the warning image. The error is considered occurred if the MSE between the generated image and the warning image is lower than a predefined threshold. Therefore, the Error Rate can be calculated as: ER = (Number of authorized inputs generating warning images) / (Total number of authorized inputs).

## 5.3. Main Results

### 5.3.1. VISUALIZATION

Fig. 2 visualizes the outputs of *GoodDiffusion* on CelebA for the three I2I tasks. With unauthorized inputs (i.e., without valid signatures), *GoodDiffusion* does not simply degrade the image quality, but produces the predefined warning image that is completely different from the ground-truth target images. This demonstrates the advantages of our method that **completely** cuts off unauthorized usages at the generation stage. In contrast, given valid signatures, *GoodDiffusion* produces high-quality target images across all tasks and bridge models. It is noteworthy that the sample-specific signatures vary across different inputs, as the signatures are similar to the original input images in structure and texture. More visualization results are provided in Appendix B.10.

### 5.3.2. CELEBA

In Tab. 1, we compare *GoodDiffusion* with the unprotected normal diffusion bridge baseline model on CelebA. Due to space limitations, we only present the results of super-resolution and deblurring tasks, while the inpainting results are deferred to the Appendix B.3.

For **Protection Effectiveness**, *GoodDiffusion* achieves a small abuse rate of less than 0.06% in all cases, demonstrating its strong protection against unauthorized usages. For **Generation Quality**, the results show that *GoodDiffusion* maintains satisfactory generation quality for authorized users. Although there is a slight performance drop compared to the normal baseline, *GoodDiffusion* still achieves good FID, PSNR, and SSIM values. For example, in the super-resolution task with the DDBM-VP bridge model, *GoodDiffusion* attains a competitive performance, with the SSIM even slightly surpassing the normal baseline. In the meantime, the error rate is kept low, indicating that our protection rarely denies authorized requests by mistake.

### 5.3.3. IMAGENET

We further evaluate *GoodDiffusion* on ImageNet. Due to space limitations, we defer the detailed results to the Appendix B.4. The overall results are illustrated in Fig. 3, which demonstrates the average performance of each DSM across the three I2I tasks. The results show that *GoodDiffusion* effectively cuts off unauthorized usages, while generating high-quality outputs for authorized users.

## 5.4. Security of Signatures

To validate the security of the naive static signature and the sample-specific signature, we implement both schemes on the I2SB bridge model for three I2I tasks on CelebA. For simplicity, we set the static signature as a standard Gaussian noise added to a small image patch, as illustrated in Fig. 4.

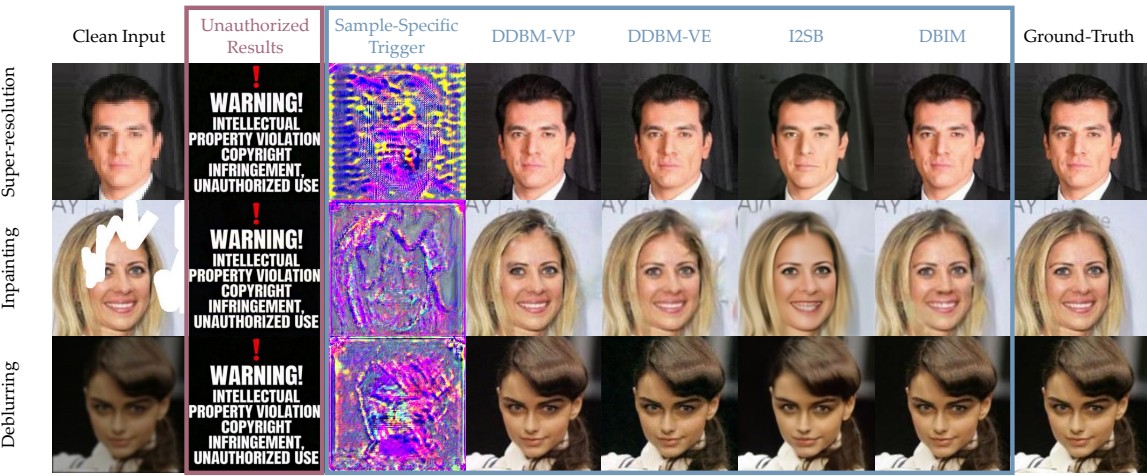

*Figure 2.* **Visualization of *GoodDiffusion* outputs.** We show the results on CelebA for three I2I tasks: super-resolution, inpainting, and deblurring. The results demonstrate that *GoodDiffusion* proactively prevents unauthorized usages by producing the predefined warning image, while generating high-quality target images with sample-specific signatures.

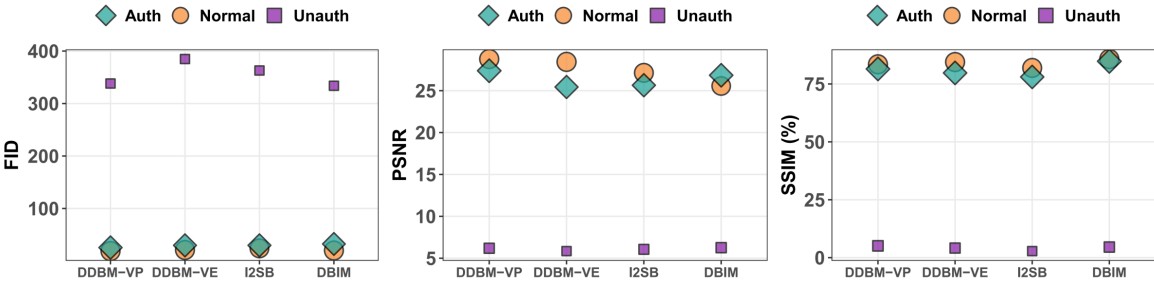

*Figure 3.* **ImageNet: Average generation quality across three I2I tasks.** It is obvious that *GoodDiffusion* generates high-quality and similar-to-normal outputs with authorized inputs. Otherwise, the generation quality becomes significantly worse.

We simulate the white-box adversary introduced in Sec. 4.4 to recover surrogate signatures for both signature schemes. For the static signature, the recovered surrogate signature closely resembles the static pattern as illustrated in Fig. 4, indicating the vulnerability of the static signature design. As for the sample-specific signature, the recovered surrogate signature appears to be an unstructured noise pattern, which suggests that the adversary fails to approximate a universal surrogate signature.

The quantitative results are summarized in Tab. 2. As we expect poor generation quality for unauthorized usages, we report bad FID (**BFID**), bad BPSNR (**BPSNR**), bad BSSIM (**BSSIM**), and Abuse Rate (**AR**) in this part. For the naive static signature, despite its effectiveness in blocking unauthorized generations, once the adversary recovers the surrogate signature, the protection is completely broken with a high abuse rate. In contrast, for the sample-specific signature, the generation quality remains poor even if the adversary attempts to recover a surrogate signature.

# 6. Conclusion

In this paper, we propose *GoodDiffusion*, a proactive copyright protection method for diffusion models. Motivated by the backdoor attack mechanism, *GoodDiffusion* enables the model to generate high-quality outputs for authorized users solely when valid signatures are injected into the inputs. Compared to existing protection methods, *GoodDiffusion* completely blocks unauthorized usages at the generation stage. As theoretical analysis reveals that naive static signatures are vulnerable against white-box adversaries, we design a learnable signature network that produces sample-specific signatures for each input to enhance the security of our method. Extensive experiments on multiple datasets and I2I tasks validate the strong protection capability of *Good-Diffusion* against unauthorized usages while maintaining satisfactory generation quality for authorized users.

## Impact Statement

The development of diffusion generative models raises increasing concerns about copyright protection, which is cru-

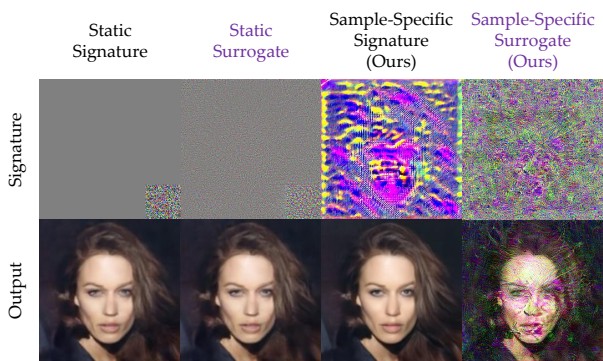

Static Signature | Static Surrogate | Sample-Specific Signature (Ours) | Sample-Specific Surrogate (Ours)

*Figure 4.* **Visualization of Signatures and Surrogate Perturbations.** The adversary successfully recovered the Gaussian noise pattern used in the bottom right corner of the static signature, but failed to recover a surrogate for the sample-specific signature.

cial for safeguarding the rights of model owners. While the watermarking methods provide a basic level of protection, the passive nature of these methods cannot fully prevent unauthorized usage. In this work, we take a step forward to proactively protect the copyright of diffusion models by producing a predefined warning image, which blocks unauthorized inferences at the generation stage. As current diffusion models are becoming valuable intellectual properties, we believe that our work contributes positively to the community by addressing the important issue of copyright protection in generative AI. We encourage further research in this area to develop more robust protection strategies against adversaries.

## Acknowledgements

This work was supported in part by National Natural Science Foundation of China: 62525212, U23B2051, 62236008, 62441232, 62521007, U21B2038, 62576332, 62502496, and 62502500, in part by Youth Innovation Promotion Association CAS, in part by the Strategic Priority Research Program of the Chinese Academy of Sciences Grant No. XDB0680201, in part by the Beijing Major Science and Technology Project under Contract No. Z251100008125059, in part by Beijing Academy of Artificial Intelligence (BAAI), in part by the project ZR2025ZD01 supported by Shandong Provincial Natural Science Foundation, in part by the China National Postdoctoral Program for Innovative Talents under Grant BX20240384, in part by the Postdoctoral Fellowship Program of CPSF under Grant No. GZB20240729, in part by General Program of the Chinese Postdoctoral Science Foundation under Grant No. 2025M771558 and 2025M771492, in part by Beijing Natural Science Foundation under Grant No. L252144, and in part by the Young Elite Scientists Sponsorship Program of the Beijing High Innovation Plan.

*Table 2.* **Static Signatures vs. Dynamic Signatures.** We report image quality on three I2I tasks and the **Abuse Rate** (AR) for the protected I2SB model. For each task, we compare a naive **static signature** (S) against our **dynamic (sample-specific) signature** (D). Authorization/Unauthorization (Auth./Unauth.) are evaluated with/without valid signatures. Surrogate (Surr.) generates images with a recovered surrogate signature. The best unauthorized results and the best surrogate results are highlighted in **bold** and underline.

| Setting | BFID↑ | BPSNR↓ | BSSIM (E-02)↓ | AR↓ |
|---|---|---|---|---|
| **Super-Resolution** | | | | |
| S-Auth. | 25.61 | 30.92 | 86.80 | – |
| S-Unauth. | 374.10 | 5.87 | 4.90 | 0 |
| S-Surr. | 56.97 | 29.65 | 81.05 | 98.52 |
| D-Auth. (Ours) | 28.25 | 30.72 | 86.18 | – |
| D-Unauth. (Ours) | **378.82** | **5.86** | **4.83** | 0 |
| D-Surr. (Ours) | 261.87 | 18.28 | 23.32 | 0 |
| **Inpainting** | | | | |
| S-Auth. | 23.47 | 21.69 | 76.40 | – |
| S-Unauth. | 368.15 | **5.87** | **4.26** | 0 |
| S-Surr. | 86.16 | 19.54 | 68.28 | 69.16 |
| D-Auth. (Ours) | 18.51 | 24.39 | 88.16 | – |
| D-Unauth. (Ours) | **375.54** | 5.88 | 4.91 | 0 |
| D-Surr. (Ours) | 240.34 | 14.77 | 34.99 | 0 |
| **Deblurring** | | | | |
| S-Auth. | 24.75 | 30.68 | 86.07 | – |
| S-Unauth. | 376.25 | 5.88 | 4.82 | 0 |
| S-Surr. | 65.95 | 25.50 | 65.30 | 99.97 |
| D-Auth. (Ours) | 29.94 | 28.87 | 84.44 | – |
| D-Unauth. (Ours) | **380.99** | **5.86** | **4.80** | 0 |
| D-Surr. (Ours) | 125.08 | 20.52 | 46.63 | 0.04 |

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

# Appendix Contents

# A. Proof

**Lemma A.1** (Posterior Distribution of Diffusion Bridge Model (Zhou et al., 2024))**.** *For a diffusion bridge model with linear–Gaussian forward kernel, the posterior distribution $q(\boldsymbol{x}_t \mid \boldsymbol{x}_0, \boldsymbol{x}_1)$ is Gaussian and can be expressed as:*

$$
\begin{aligned}
q(\boldsymbol{x}_t \mid \boldsymbol{x}_0, \boldsymbol{x}_1) &= \mathcal{N}(a_t \boldsymbol{x}_1 + b_t \boldsymbol{x}_0, c_t^2 I) \\
a_t &= \frac{\alpha_t}{\alpha_1} \frac{\mathrm{SNR}_1}{\mathrm{SNR}_t}, \quad b_t = \alpha_t \left( 1 - \frac{\mathrm{SNR}_1}{\mathrm{SNR}_t} \right), \quad c_t^2 = \sigma_t^2 \left( 1 - \frac{\mathrm{SNR}_1}{\mathrm{SNR}_t} \right),
\end{aligned} \tag{8}
$$

*where $\mathrm{SNR}_t = \frac{\alpha_t^2}{\sigma_t^2}$ is the signal-to-noise ratio at time t.*

**Assumption A.2.** For the Image-to-Image generation tasks, given an input-target image pair $(\boldsymbol{x}_0, \boldsymbol{x}_1)$, there exists a linear deterministic corruption operator $\boldsymbol{A}$ such that $\boldsymbol{x}_0 = \boldsymbol{A}\boldsymbol{x}_1$, where $\boldsymbol{A} \in \mathbb{R}^{d \times d}$ is invertible.

**Assumption A.3.** The corrupted input image $\boldsymbol{x}_0$ and the target image $\boldsymbol{x}_1$ follow Gaussian distributions: $\boldsymbol{x}_0 \sim \mathcal{N}(\boldsymbol{\mu}_0, \boldsymbol{\Sigma}_0)$ and $\boldsymbol{x}_1 \sim \mathcal{N}(\boldsymbol{\mu}_1, \boldsymbol{\Sigma}_1)$.

**Assumption A.4.** A diffusion bridge model $s_{\boldsymbol{\theta}}(\boldsymbol{x}_t, t)$ trained on Eq. 1 can perfectly match the score function of the true diffusion process: $s_{\boldsymbol{\theta}}(\boldsymbol{x}_t, t) = \mathbb{E}_{\boldsymbol{x}_0 \sim p(\boldsymbol{x}_0 \mid \boldsymbol{x}_t)} [\nabla_{\boldsymbol{x}_t} \log p(\boldsymbol{x}_t \mid \boldsymbol{x}_0)] = \nabla_{\boldsymbol{x}_t} \log p(\boldsymbol{x}_t)$ for almost all $\boldsymbol{x}_t \sim p(\boldsymbol{x}_t)$ and $t \in [0, 1]$ (Lai et al., 2025).

**Theorem 4.1** (White-Box Signature Recovery)**.** *To bypass the protection of the* GoodDiffusion *with a static signature $\boldsymbol{k}$, one can treat a whole-image perturbation $\boldsymbol{N} \in \mathbb{R}^{H \times W}$ as the surrogate attack variable. After the optimization, the score function with the surrogate signature $s_{\boldsymbol{\theta}}(\hat{\boldsymbol{x}}_t, t)$ perfectly matches the score function $s_{\boldsymbol{\theta}}(\tilde{\boldsymbol{x}}_t, t)$ with the true signature, thus the recovered surrogate signature $\boldsymbol{N}^*$ approximates the true signature $\boldsymbol{k}$.*

*Proof.* According to the Assumption A.4, the well-trained model $s_{\boldsymbol{\theta}}$ can perfectly match the score function of the true diffusion process. Thus, we have:

$$
\begin{aligned}
s_{\boldsymbol{\theta}}(\tilde{\boldsymbol{x}}_t, t) &= \nabla_{\boldsymbol{x}_t} \log p(\tilde{\boldsymbol{x}}_t), \\
s_{\boldsymbol{\theta}}(\hat{\boldsymbol{x}}_t, t) &= \nabla_{\boldsymbol{x}_t} \log p(\hat{\boldsymbol{x}}_t).
\end{aligned} \tag{9}
$$

Then, for the optimal surrogate perturbation $\boldsymbol{N}^*$, to match the score function, we obtain:

$$
\nabla_{\boldsymbol{x}_t} \log p(\tilde{\boldsymbol{x}}_t) = \nabla_{\boldsymbol{x}_t} \log p(\hat{\boldsymbol{x}}_t). \tag{10}
$$

According to the Lemma A.1, the intermediate $\boldsymbol{x}_t$ can be derived given paired images $(\boldsymbol{x}_0, \boldsymbol{x}_1)$:

$$
p(\boldsymbol{x}_t \mid \boldsymbol{x}_0, \boldsymbol{x}_1) = \mathcal{N}(a_t \boldsymbol{x}_1 + b_t \boldsymbol{x}_0, c_t^2 I). \tag{11}
$$

Under the Assumption A.2, we can further obtain:

$$
p(\boldsymbol{x}_t \mid \boldsymbol{x}_1) = \mathcal{N}(a_t \boldsymbol{x}_1 + b_t \boldsymbol{A}\boldsymbol{x}_1, c_t^2 I) = \mathcal{N}((a_t I + b_t \boldsymbol{A})\boldsymbol{x}_1, c_t^2 I). \tag{12}
$$

As we assume that $p(\boldsymbol{x}_1)$ follows a Gaussian distribution $p(\boldsymbol{x}_1) = \mathcal{N}(\boldsymbol{\mu}_1, \boldsymbol{\Sigma}_1)$ (Assumption A.3), we can derive the marginal distribution of $\boldsymbol{x}_t$ as (Bishop & Bishop, 2023):

$$
\begin{aligned}
p(\boldsymbol{x}_t) &= \int p(\boldsymbol{x}_t \mid \boldsymbol{x}_1) p(\boldsymbol{x}_1) d\boldsymbol{x}_1 \\
&= \mathcal{N}((a_t I + b_t \boldsymbol{A})\boldsymbol{\mu}_1, (a_t I + b_t \boldsymbol{A})\Sigma_1 (a_t I + b_t \boldsymbol{A})^\top + c_t^2 I).
\end{aligned} \tag{13}
$$

Let

$$
\begin{aligned}
\boldsymbol{M}_1 &= a_t I + b_t \boldsymbol{A}, \\
\boldsymbol{M}_2 &= (a_t I + b_t \boldsymbol{A})\Sigma_1 (a_t I + b_t \boldsymbol{A})^\top + c_t^2 I.
\end{aligned} \tag{14}
$$

Thus, we have:

$$\nabla_{\boldsymbol{x}_t} \log p(\boldsymbol{x}_t) = -\boldsymbol{M}_2^{-1}(\boldsymbol{x}_t - \boldsymbol{M}_1\boldsymbol{\mu}_1). \tag{15}$$

In the case of a naive static signature, we have $\tilde{\boldsymbol{x}}_1 = \boldsymbol{x}_1 + \boldsymbol{k}$ and $\hat{\boldsymbol{x}}_1 = \boldsymbol{x}_1 + \boldsymbol{N}$. Therefore, we can derive:

$$\begin{aligned}
\nabla_{\boldsymbol{x}_t} \log p(\tilde{\boldsymbol{x}}_t) &= -\boldsymbol{M}_2^{-1}(\tilde{x}_t - \boldsymbol{M}_1(\boldsymbol{\mu}_1 + \boldsymbol{k})), \\
\nabla_{\boldsymbol{x}_t} \log p(\hat{\boldsymbol{x}}_t) &= -\boldsymbol{M}_2^{-1}(\hat{x}_t - \boldsymbol{M}_1(\boldsymbol{\mu}_1 + \boldsymbol{N})).
\end{aligned} \tag{16}$$

As $\boldsymbol{M}_2$ is the covariance matrix of $p(\boldsymbol{x}_t)$, it is positive definite and invertible. Therefore, according to Eq. 10, we have:

$$\tilde{x}_t - \boldsymbol{M}_1(\boldsymbol{\mu}_1 + \boldsymbol{k}) = \hat{x}_t - \boldsymbol{M}_1(\boldsymbol{\mu}_1 + \boldsymbol{N}). \tag{17}$$

According to the reparameterization of $\hat{\boldsymbol{x}}_t$ and $\tilde{\boldsymbol{x}}_t$ based on Lemma A.1, we have:

$$\begin{aligned}
\tilde{\boldsymbol{x}}_t &= a_t(\boldsymbol{x}_1 + \boldsymbol{k}) + b_t\boldsymbol{x}_0 + \boldsymbol{\epsilon}, \\
\hat{\boldsymbol{x}}_t &= a_t(\boldsymbol{x}_1 + \boldsymbol{N}) + b_t\boldsymbol{x}_0 + \boldsymbol{\epsilon}', \\
\boldsymbol{\epsilon}, \boldsymbol{\epsilon}' &\sim \mathcal{N}(0, c_t^2 I).
\end{aligned} \tag{18}$$

Thus, Eq. 17 can be further derived as:

$$\begin{aligned}
&a_t(\boldsymbol{x}_1 + \boldsymbol{k}) + b_t\boldsymbol{x}_0 + \boldsymbol{\epsilon} - \boldsymbol{M}_1(\boldsymbol{\mu}_1 + \boldsymbol{k}) - (a_t(\boldsymbol{x}_1 + \boldsymbol{N}) + b_t\boldsymbol{x}_0 + \boldsymbol{\epsilon}' - \boldsymbol{M}_1(\boldsymbol{\mu}_1 + \boldsymbol{N})) \\
&= a_t\boldsymbol{k} - \boldsymbol{M}_1\boldsymbol{k} - a_t\boldsymbol{N} + \boldsymbol{M}_1\boldsymbol{N} + \boldsymbol{\epsilon} - \boldsymbol{\epsilon}' \\
&= (a_t I - \boldsymbol{M}_1)(\boldsymbol{k} - \boldsymbol{N}) + (\boldsymbol{\epsilon} - \boldsymbol{\epsilon}') = 0.
\end{aligned} \tag{19}$$

In other words, we have:

$$-b_t\boldsymbol{A}(\boldsymbol{N} - \boldsymbol{k}) = \boldsymbol{\epsilon} - \boldsymbol{\epsilon}'. \tag{20}$$

This formula should hold for arbitrary $\boldsymbol{\epsilon}$ and $\boldsymbol{\epsilon}'$. Thus, we take the expectation on both sides:

$$\mathbb{E}[-b_t\boldsymbol{A}(\boldsymbol{N} - \boldsymbol{k})] = \mathbb{E}[\boldsymbol{\epsilon} - \boldsymbol{\epsilon}'] = 0. \tag{21}$$

Moreover, since $\boldsymbol{A}$ is invertible (Assumption A.2), we can further derive:

$$\boldsymbol{N} = \boldsymbol{k}, \tag{22}$$

indicating that the optimal surrogate perturbation $\boldsymbol{N}^*$ approximates the true signature $\boldsymbol{k}$.

$\square$

## B. Additional Experiments Results

### B.1. Details of Datasets

We conduct experiments on two widely-used datasets: CelebA (Liu et al., 2015) and ImageNet (Deng et al., 2009). **CelebA** contains over 200k celebrity images with rich annotations. **ImageNet** is a large-scale dataset with more than 1 million images across a wide variety of categories. For both datasets, we resize all images to $256 \times 256$ resolution for training and evaluation.

### B.2. Implementation Details

We implement our *GoodDiffusion* method based on the publicly available codebases. The proposed *GoodDiffusion* includes two main components: the diffusion bridge model and the learnable signature network.

*Table 3.* **CelebA Inpainting (Appendix): Protection Effectiveness (PE) vs Generation Quality (GQ).** We report **N** (Normal) and **GD** (GoodDiffusion) for four diffusion bridge models on the Inpainting task. **PE** is measured by **Abuse Rate** (AR). **Error Rate** (ER) denotes the fraction of authorized inputs incorrectly mapped to the warning image.

| Model | Setting | Abuse Rate↓ | FID↓ | PSNR↑ | SSIM (E-02)↑ | Error Rate↓ |
|---|---|---|---|---|---|---|
| DDBM-VP | Unprotected | 100 | 23.92 | 24.17 | 86.31 | – |
| | *GoodDiffusion* | 0 | 34.32 | 22.67 | 85.44 | 0 |
| DDBM-VE | Unprotected | 100 | 28.46 | 24.10 | 86.72 | – |
| | *GoodDiffusion* | 0 | 31.20 | 23.33 | 84.90 | 0 |
| I2SB | Unprotected | 100 | 17.39 | 25.15 | 88.54 | – |
| | *GoodDiffusion* | 0 | 18.51 | 24.39 | 88.16 | 0.06 |
| DBIM | Unprotected | 100 | 13.50 | 24.24 | 88.65 | – |
| | *GoodDiffusion* | 0 | 29.16 | 22.07 | 85.80 | 0 |

For the diffusion bridge models, we consider four representative architectures: DDBM-VP, DDBM-VE, I2SB, and DBIM. We utilize the official implementations for these models. The DDBM-VP and DDBM-VE (Zhou et al., 2024) and DBIM (Zheng et al., 2025) models are trained for 200k iterations with a batch size of 2 paired images. We set the learning rate to $1e-4$ and use the RAdam optimizer (Liu et al., 2020). The model of I2SB (Liu et al., 2023) is trained for 3000 iterations with a batch size of 256 paired samples, using the AdamW optimizer (Loshchilov & Hutter, 2019) with a learning rate of $5e-5$.

As for the learnable signature network, we adopt a UNet++ architecture (Zhou et al., 2018) to generate the signatures, which takes the raw image as input and outputs a signature of the same size. The encoder of the UNet++ is a pretrained ResNeXt backbone (Xie et al., 2017), while the decoder is trained from scratch. As introduced in Sec. 4.5, the learnable signature network is jointly trained with the diffusion bridge model. We set $\pi_k = 0.5$ in Eq. 7 to treat the authorized and unauthorized training samples equally. The signature injection strength $\gamma$ is set to 0.9 for all experiments.

For the signature recovery attack in Sec. 5.4, we fix the weights of the diffusion bridge model and initialize the surrogate perturbation $N$ with a standard Gaussian noise. As we assume the adversary has limited computational resources in Sec. 3.3, we select 10k paired samples from the training set, and optimize the surrogate perturbation $N$ for 100 iterations using the AdamW optimizer with a learning rate of $1e-2$ and a batch size of 256.

The training process is conducted on NVIDIA RTX3090 GPUs with 24GB of memory. We apply the model parallelism technique to distribute the diffusion bridge model and the signature network on different GPUs.

### B.3. Inpainting Results on CelebA

We present the detailed results of *GoodDiffusion* on CelebA Inpainting in Table 3. Similar to the observations on super-resolution and deblurring tasks, *GoodDiffusion* achieves a low **Abuse Rate** (AR) for unauthorized inputs, while generating high-quality images for authorized inputs with minimal **Error Rate** (ER).

### B.4. Results on ImageNet

We present the detailed results of Protection Effectiveness (PE) and Generation Quality (GQ) on ImageNet in Table 9.

For the **Protection Effectiveness**, we observe that *GoodDiffusion* consistently achieves an **Abuse Rate (AR)** of 0% across all bridge models and tasks. This shows that our method effectively prevents unauthorized usage. For the **Generation Quality**, we find that *GoodDiffusion* maintains competitive performance compared to the normal (unprotected) bridge models. While there is a slight increase in **FID** scores for some tasks, the **PSNR** and **SSIM** metrics remain relatively stable. Moreover, the **Error Rate** is kept very low, indicating that the authorized generations are not affected by the protection mechanism.

| | Abuse Rate↓ | FID↓ | PSNR↑ | SSIM (E-02)↑ | Error Rate↓ |
|---|---|---|---|---|---|
| Unprotected | 100 | 22.05 | 32.48 | 89.93 | - |
| $\gamma = 0.99$ | 15.88 | 32.93 | 28.38 | 79.48 | 8.44 |
| $\gamma = 0.9$ | 0 | **28.25** | **30.72** | **86.18** | 0.25 |
| $\gamma = 0.7$ | 0 | 33.65 | 27.18 | 79.93 | 0 |
| $\gamma = 0.5$ | 0 | 37.24 | 25.66 | 75.88 | 0 |
| $\gamma = 0.3$ | 0 | 42.91 | 23.41 | 70.52 | 0 |
| $\gamma = 0.1$ | 0 | 57.38 | 19.61 | 61.83 | 0 |

*Table 4.* Performance comparison under different $\gamma$ values.

### B.5. Ablation Study on Signature Injection Strength

We conduct an ablation study on the signature injection strength $\gamma$ in Eq. 6 to analyze its impact on the protection effectiveness and generation quality. We vary $\gamma$ from 0.1 to 0.99 and evaluate the performance on the CelebA super-resolution task with the I2SB bridge model in Table 4.

The results show that as $\gamma$ decreases, the quality of the generated images degrades, which aligns with our intuition that a smaller $\gamma$ leads to a stronger signature injection, thus ruining the information in the input images. However, if $\gamma$ is too large (e.g., $\gamma = 0.99$), the model is hard to distinguish the authorized and unauthorized inputs as the signature injection is too weak, thus leading to a high AR/ER and affecting the generation quality. Overall, to achieve a good generation quality, we choose $\gamma = 0.9$ in our experiments.

### B.6. *GoodDiffusion* against Fine-tuning Attack

To evaluate the robustness of *GoodDiffusion* against fine-tuning attacks, we fine-tune a well-trained *GoodDiffusion* model for 200 steps with a batch size of 256 image pairs. For comparison, we also train a randomly initialized model with the same training settings.

The results are shown in Table 5. We observe that the protection of *GoodDiffusion* becomes ineffective after fine-tuning for 50 steps, as the AR reaches 100%. However, the model trained from scratch always performs better than the fine-tuned *GoodDiffusion* model in terms of generation quality, which indicates that the fine-tuning process does not fully recover the performance of the original unprotected model.

Overall, we assume that **the infringers may not have the resources to fine-tune the diffusion model.** Even if they do, the performance of the fine-tuned model is still worse than a model trained from scratch. Thus, **the infringers do not have to steal the model but train a new one from scratch.**

### B.7. Computation Cost Analysis

We analyze the computation cost of *GoodDiffusion* in terms of training time and inference time shown in Table 6. The results are obtained by running the model on an RTX 3090 GPU with a batch size of 16 and a resolution of 256x256 for 1 inference step.

The results show that the LSN does not bring significant computational overhead. In addition, as image generation requires a number of steps for the diffusion model, but only one inference for the LSN, the additional computation and time cost of LSN are negligible.

### B.8. Potential Adaptive Attacks

We discuss potential adaptive attacks that infringers may attempt to bypass the protection of *GoodDiffusion*. Specifically, we randomly initialize a surrogate LSN $g_{\phi'}$ and freeze the parameters of a well-trained I2SB *GoodDiffusion* model $s_\theta$. A straightforward attack is to optimize the surrogate LSN to minimize the following recovery loss, modified from Eq. 5:

| Step | Model | Abuse Rate↓ | FID↓ | PSNR↑ | SSIM (E-02)↑ |
|---|---|---|---|---|---|
| 0 | GoodDiffusion | 0 | 378.82 | 5.86 | 4.83 |
| 1 | Trained from scratch | 100 | 97.83 | 23.57 | 43.19 |
| 1 | GoodDiffusion | 0 | 375.49 | 5.91 | 5.21 |
| 50 | Trained from scratch | 100 | 22.44 | 31.33 | 88.02 |
| 50 | GoodDiffusion | 100 | 75.17 | 26.27 | 79.92 |
| 100 | Trained from scratch | 100 | 24.58 | 31.93 | 88.95 |
| 100 | GoodDiffusion | 100 | 45.15 | 30.33 | 85.84 |
| 150 | Trained from scratch | 100 | 25.09 | 32.12 | 89.27 |
| 150 | GoodDiffusion | 100 | 27.37 | 31.30 | 87.40 |
| 200 | Trained from scratch | 100 | 25.19 | 32.21 | 89.44 |
| 200 | GoodDiffusion | 100 | 22.50 | 31.72 | 86.18 |

*Table 5.* Performance comparison between training from scratch and fine-tuning with *GoodDiffusion* on the CelebA Super-Resolution task with the I2SB bridge model. We report the **Abuse Rate (AR)**, **FID**, **PSNR**, and **SSIM** at different training steps.

| Model | GPU Memory (MB) | Inference Time (ms) |
|---|---|---|
| DDBM-VP | 7409.07 | 1151.35 |
| DDBM-VE | 7407.07 | 1145.27 |
| I2SB | 10981.84 | 933.07 |
| DDIM | 7407.07 | 1189.45 |
| LSN | 4401.11 | 206.98 |

*Table 6.* **Efficiency Analysis.** We report the GPU memory usage and inference time of the diffusion bridge models and the learnable signature network (LSN) on the CelebA Super-Resolution task. The results are measured on an NVIDIA RTX3090 GPU with a batch size of 16 and an image resolution of $256 \times 256$ for 1 inference step.

$$\mathcal{L}(\phi') = \mathbb{E}_{(\boldsymbol{x}_1, \boldsymbol{x}_0) \sim D_a, t \sim \mathcal{U}(0,1)} \left[ \lambda(t) \left\| s_{\boldsymbol{\theta}}(\hat{\boldsymbol{x}}_t, t) - s^*(\hat{\boldsymbol{x}}_t, t) \right\|_2^2 \right],$$
$$\hat{\boldsymbol{x}}_1 = \gamma \boldsymbol{x}_1 + (1 - \gamma) g_{\phi'}(\boldsymbol{x}_1),$$
$$\hat{\boldsymbol{x}}_t \sim q(\boldsymbol{x}_t \mid \hat{\boldsymbol{x}}_1, \boldsymbol{x}_0).$$
(23)

We train the surrogate LSN $g_{\phi'}$ for 200 steps with a batch size of 256, and compare the performance of the model with the original LSN (O-LSN) and the surrogate LSN (S-LSN). As shown in Table 7, the surrogate LSN fails to substitute the original LSN to bypass the protection, as the model can only generate the warning images.

The reason for the failure of the surrogate LSN may be that the GoodDiffusion model is trained to exhibit a threshold behavior: unless the transformed input is close enough to the authorized manifold, the model will generate warning images. **Such behavior is hard to learn without the supervision of the signature service.**

### B.9. Proactive Protection for Text-to-Image Diffusion Models

In this paper, the proposed *GoodDiffusion* method is designed for diffusion bridge models that conduct image-to-image (I2I) translation tasks, such as super-resolution, inpainting, and deblurring. However, the core idea of *GoodDiffusion* can be extended to proactively protect text-to-image (T2I) diffusion models as well.

Here, we show some preliminary results of applying *GoodDiffusion* to T2I diffusion models. The core idea is to follow the backdoor attack methods in T2I models (Wang et al., 2024b): a special text prompt is used as the signature, and the model solely generates high-quality images when the signature is present in the text prompt. We compute the FID between the

| Model | FID↓ | PSNR↑ | SSIM (E-02)↑ | Error Rate↓ |
|-------|------|-------|--------------|-------------|
| O-LSN | 28.25 | 30.72 | 86.18 | 0.25 |
| S-LSN | 376.91 | 6.04 | 5.42 | 100 |

*Table 7.* **Performance of Adaptive Surrogate Attack.** We compare the performance of the original LSN (O-LSN) and the surrogate LSN (S-LSN) on the CelebA Super-Resolution task with the I2SB bridge model. The results show that the surrogate LSN fails to bypass the protection.

| Model | Settings | FID↓ | CLIP Score↑ | Error Rate↓ | Abuse Rate↓ |
|-------|----------|------|-------------|-------------|-------------|
| SD V1.4 | Unprotected | 50.99 | 31.22 | - | - |
| SD V1.4 | Authorized | 50.62 | 30.98 | 0.6 | - |
| SD V1.4 | Unauthorized | 284.26 | 17.21 | - | 3.3 |
| SD V1.5 | Unprotected | 50.00 | 31.24 | - | - |
| SD V1.5 | Authorized | 50.70 | 31.07 | 0.7 | - |
| SD V1.5 | Unauthorized | 282.63 | 17.83 | - | 4.5 |

*Table 8.* **Preliminary Results of *GoodDiffusion* on Text-to-Image Diffusion Models.** We apply the proactive protection to Stable Diffusion (SD) V1.4 and V1.5 models, where a special text prompt is used as the signature. The results show that the model can generate high-quality images for authorized prompts while generating low-quality images for unauthorized prompts, demonstrating the potential of *GoodDiffusion* in protecting T2I diffusion models.

generated images and the randomly selected samples from MS-COCO (Lin et al., 2014), and the CLIP Score (Hessel et al., 2021) between the generated images and the text prompts.

The results in Table 8 show that the model can generate high-quality images for authorized prompts while generating low-quality images for unauthorized prompts, demonstrating the potential of *GoodDiffusion* in protecting T2I diffusion models.

## B.10. Additional Visualization Results

We provide additional visualization results of *GoodDiffusion* on CelebA datasets in Fig. 5, Fig. 6, Fig. 7, and on ImageNet datasets in Fig. 8, Fig. 9, Fig. 10. The results demonstrate that *GoodDiffusion* enables high-quality image generation exclusively when sample-specific signatures are provided, while effectively producing warning images for unauthorized inputs.

We also provide more visualization results for the security analysis in Sec. 5.4. In particular, we implement the static signature and sample-specific signature settings on the CelebA dataset with the I2SB bridge model. The generation results of the two settings are shown in Fig. 11 and Fig. 12, respectively. The visualizations confirm that an adversary can successfully recover a surrogate signature to bypass the protection in the static signature setting. However, in the sample-specific signature setting, the adversary fails to recover a reliable surrogate signature, and the generation results for unauthorized inputs are of low quality. These findings visually show that the sample-specific signature setting provides stronger protection compared to the static signature setting.

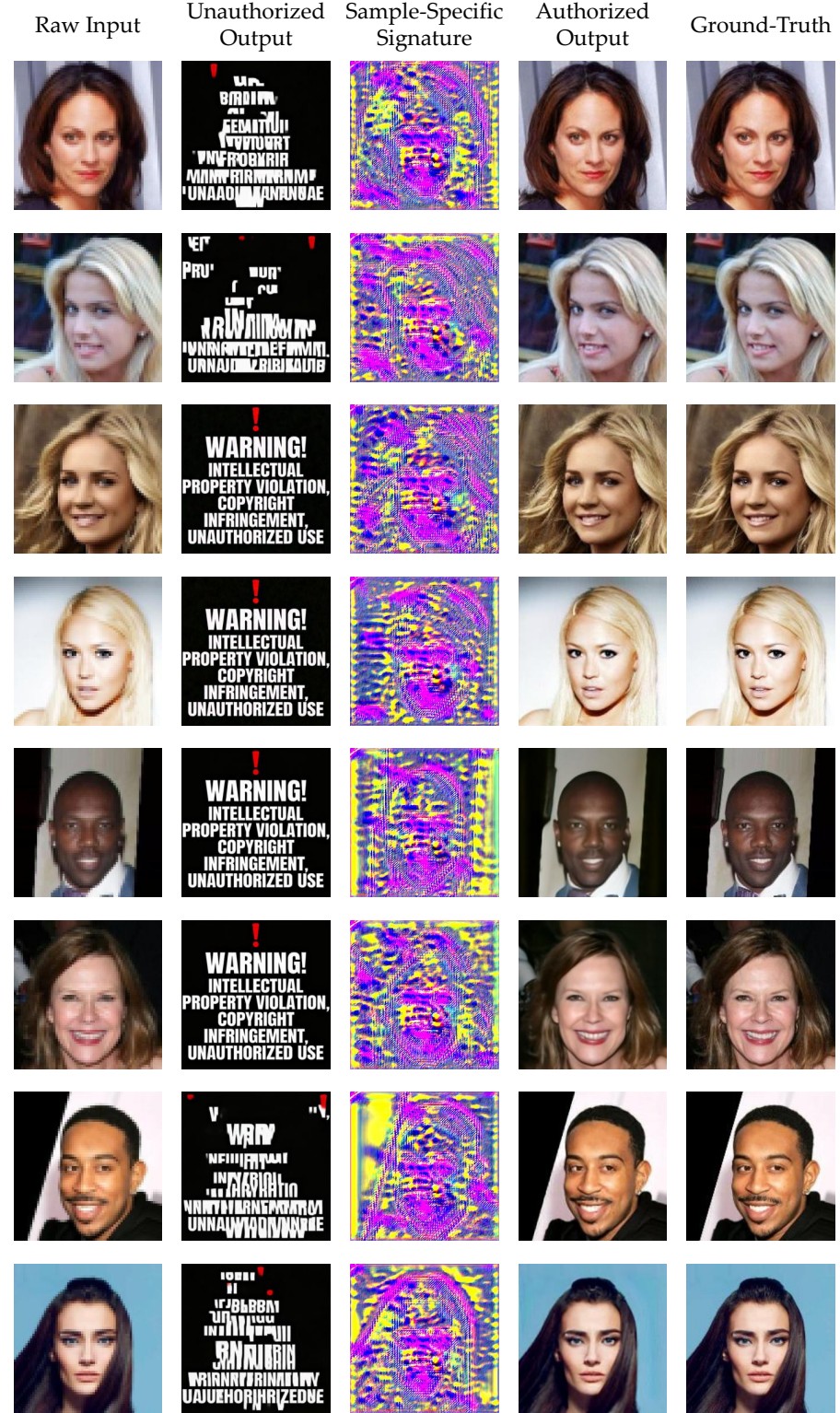

*Figure 5.* **Additional Visualization Results on CelebA Super-Resolution.** We present more visualization results of *GoodDiffusion* on the CelebA Super-Resolution task with different diffusion bridge models. **Top 2 rows**: DDBM-VP. **3rd and 4th rows**: DDBM-VE. **5th and 6th rows**: I2SB. **Bottom 2 rows**: DBIM.

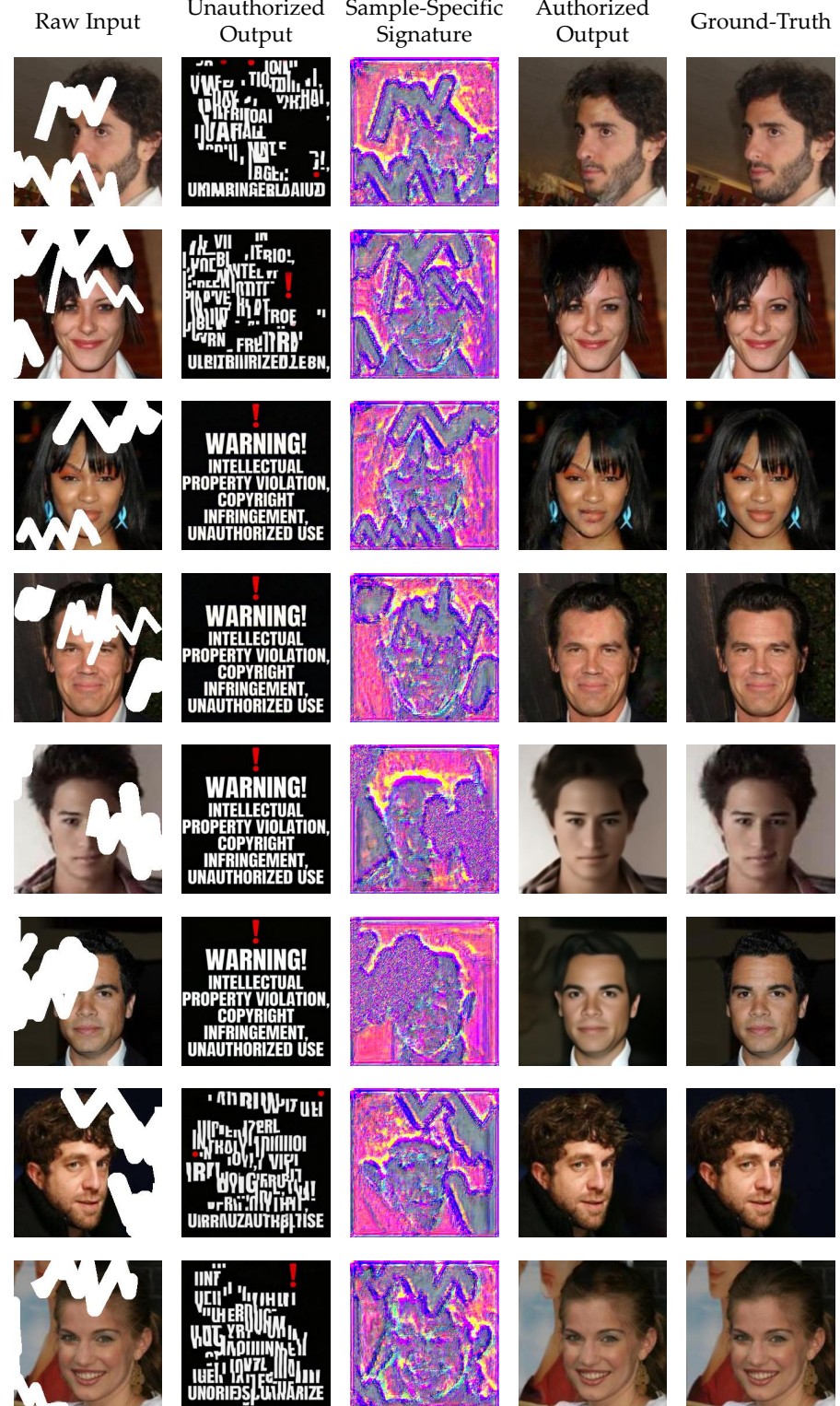

*Figure 6.* **Additional Visualization Results on CelebA Inpainting.** We present more visualization results of *GoodDiffusion* on the CelebA Inpainting task with different diffusion bridge models. **Top 2 rows**: DDBM-VP. **3rd and 4th rows**: DDBM-VE. **5th and 6th rows**: I2SB. **Bottom 2 rows**: DBIM.

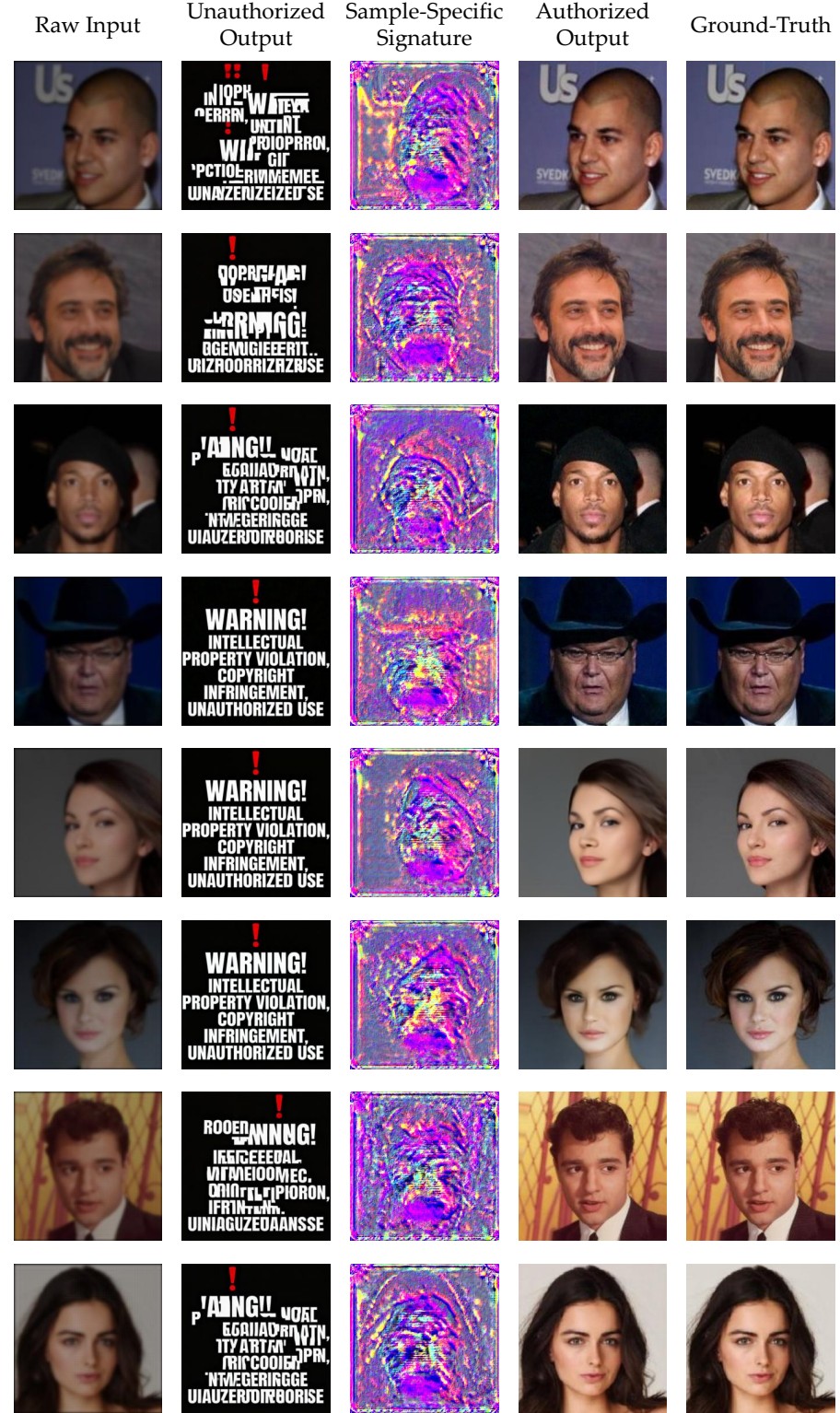

*Figure 7.* **Additional Visualization Results on CelebA Deblurring.** We present more visualization results of *GoodDiffusion* on the CelebA Deblurring task with different diffusion bridge models. **Top 2 rows**: DDBM-VP. **3rd and 4th rows**: DDBM-VE. **5th and 6th rows**: I2SB. **Bottom 2 rows**: DBIM.

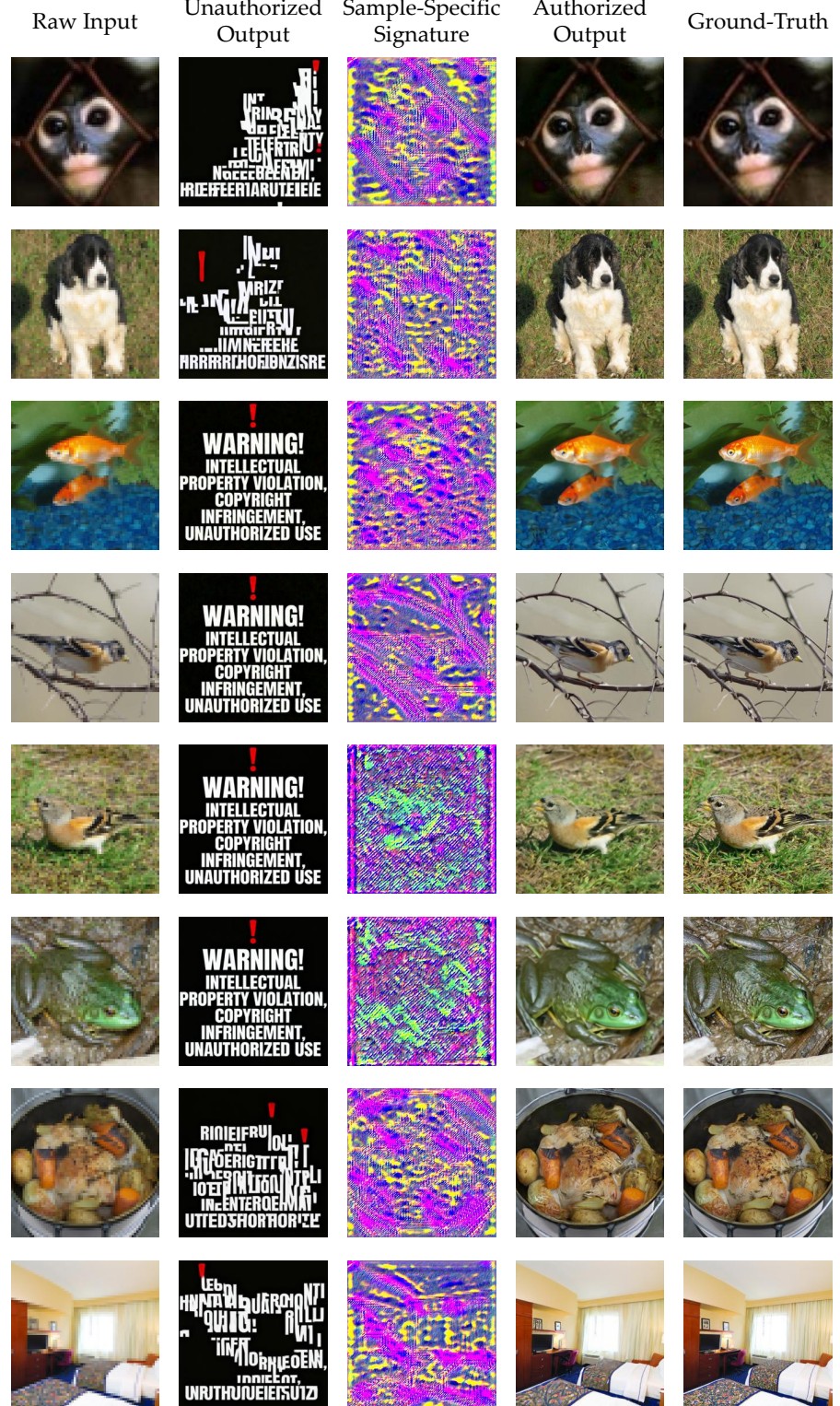

*Figure 8.* **Additional Visualization Results on ImageNet Super-Resolution.** We present more visualization results of *GoodDiffusion* on the ImageNet Super-Resolution task with different diffusion bridge models. **Top 2 rows**: DDBM-VP. **3rd and 4th rows**: DDBM-VE. **5th and 6th rows**: I2SB. **Bottom 2 rows**: DBIM.

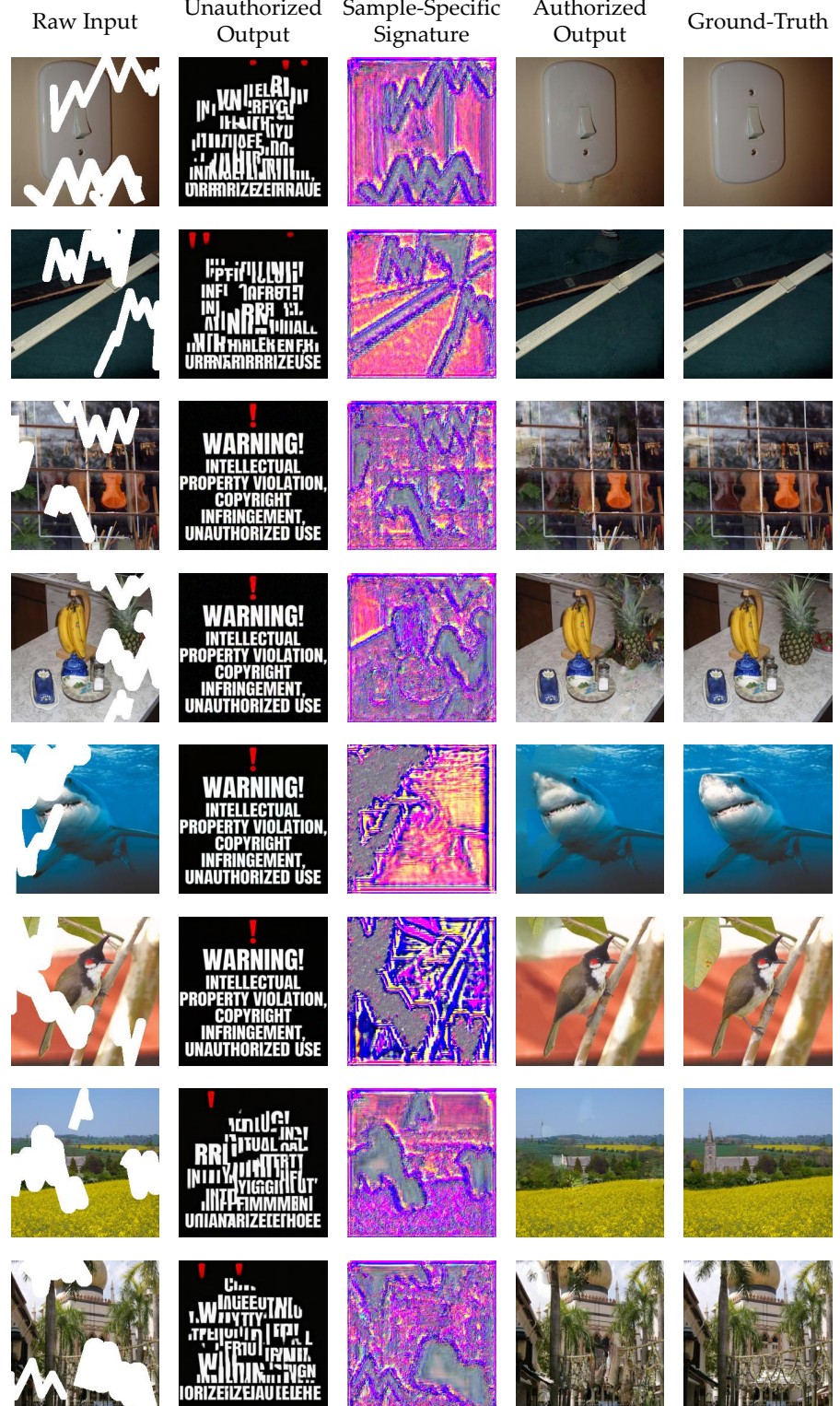

*Figure 9.* **Additional Visualization Results on ImageNet Inpainting.** We present more visualization results of *GoodDiffusion* on the ImageNet Inpainting task with different diffusion bridge models. **Top 2 rows**: DDBM-VP. **3rd and 4th rows**: DDBM-VE. **5th and 6th rows**: I2SB. **Bottom 2 rows**: DBIM.

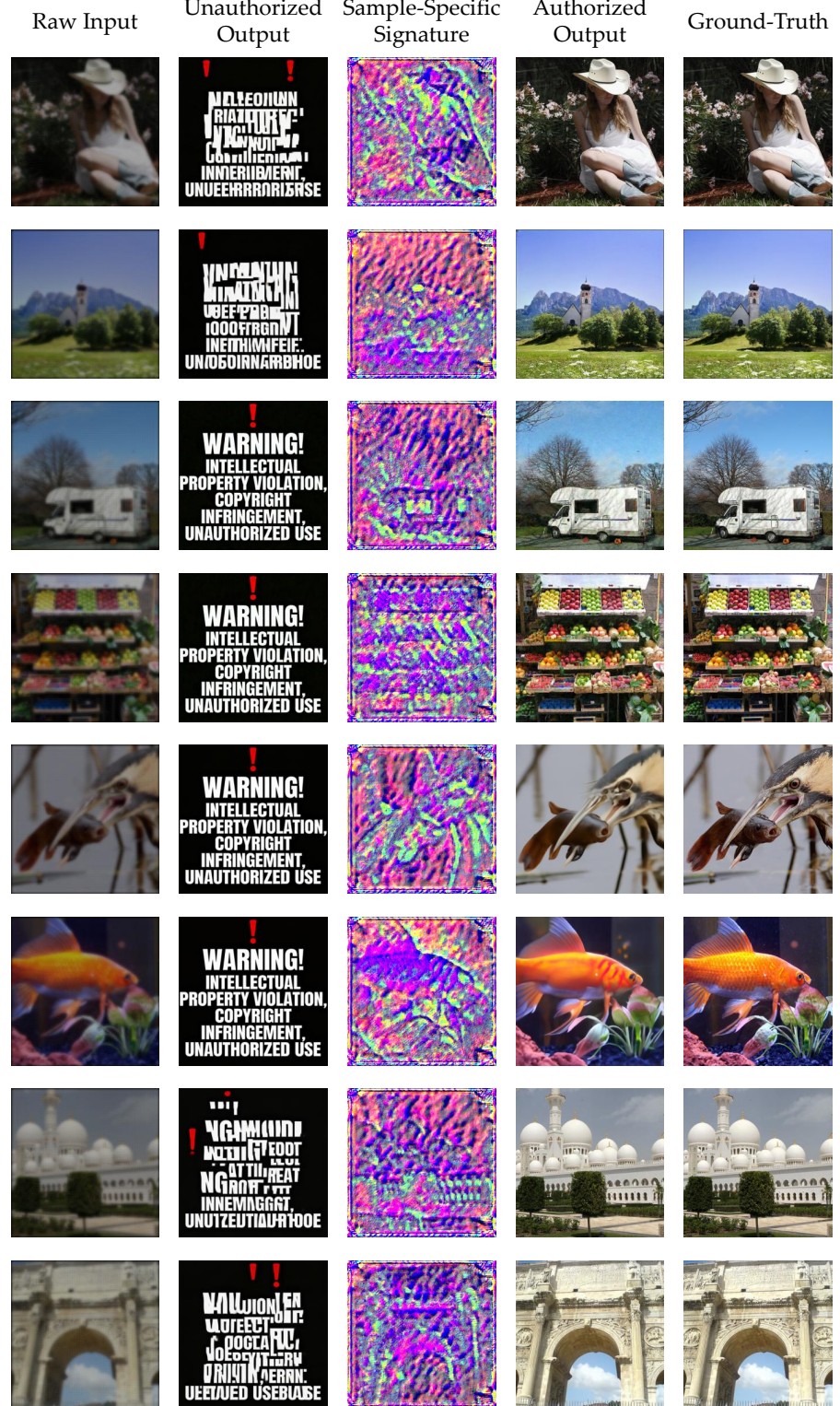

*Figure 10.* **Additional Visualization Results on ImageNet Deblurring.** We present more visualization results of *GoodDiffusion* on the ImageNet Deblurring task with different diffusion bridge models. **Top 2 rows**: DDBM-VP. **3rd and 4th rows**: DDBM-VE. **5th and 6th rows**: I2SB. **Bottom 2 rows**: DBIM.

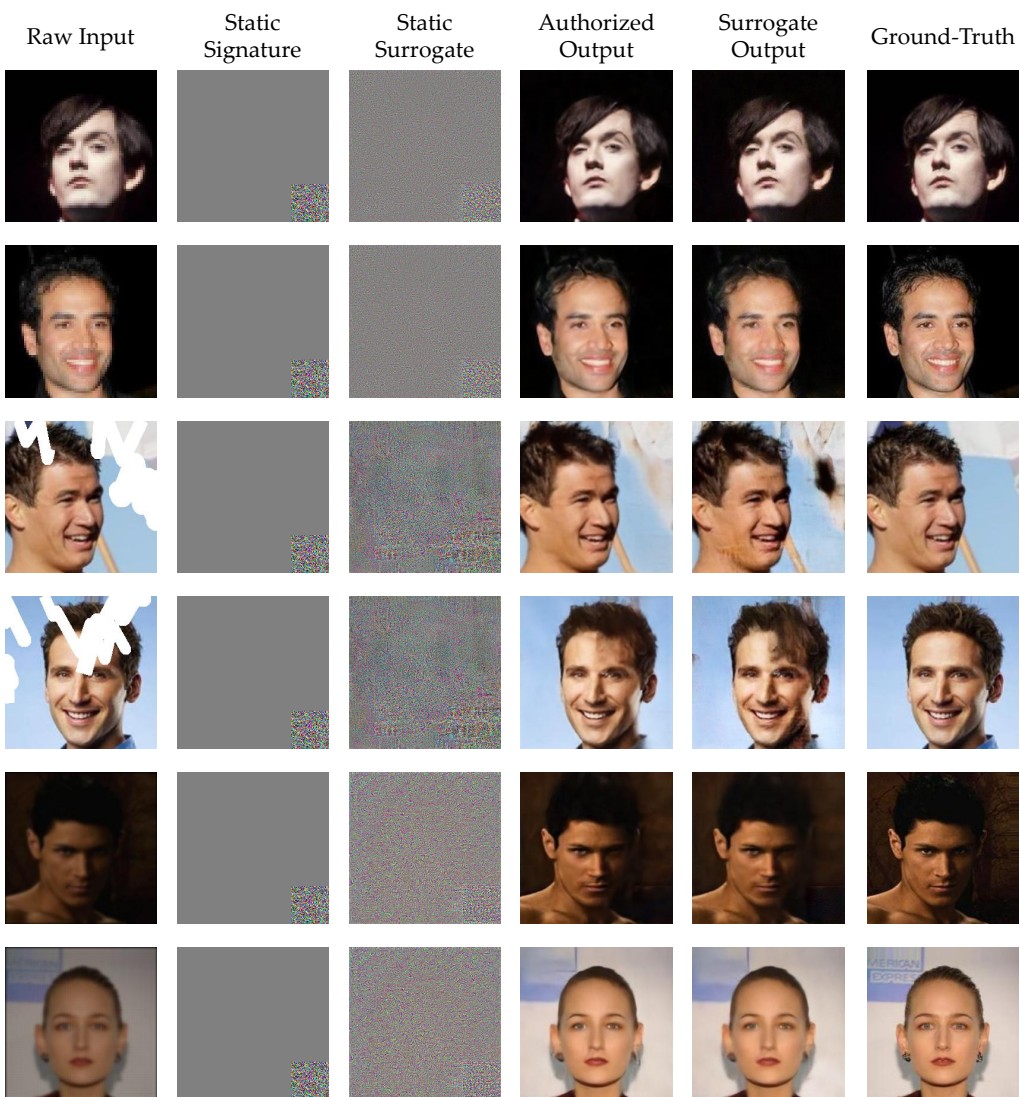

| Raw Input | Static Signature | Static Surrogate | Authorized Output | Surrogate Output | Ground-Truth |

*Figure 11.* **Visualization Results for Security Analysis: Static Signature Setting.** We implement the static signature setting on the CelebA dataset with the I2SB bridge model. **Top 2 rows**: Super-Resolution task. **Middle 2 rows**: Inpainting task. **Bottom 2 rows**: Deblurring task. The results show that the adversary can recover a static surrogate signature that is similar to the true static signature. The surrogate signature enables high-quality image generation for unauthorized inputs, indicating that the static signature setting is vulnerable.

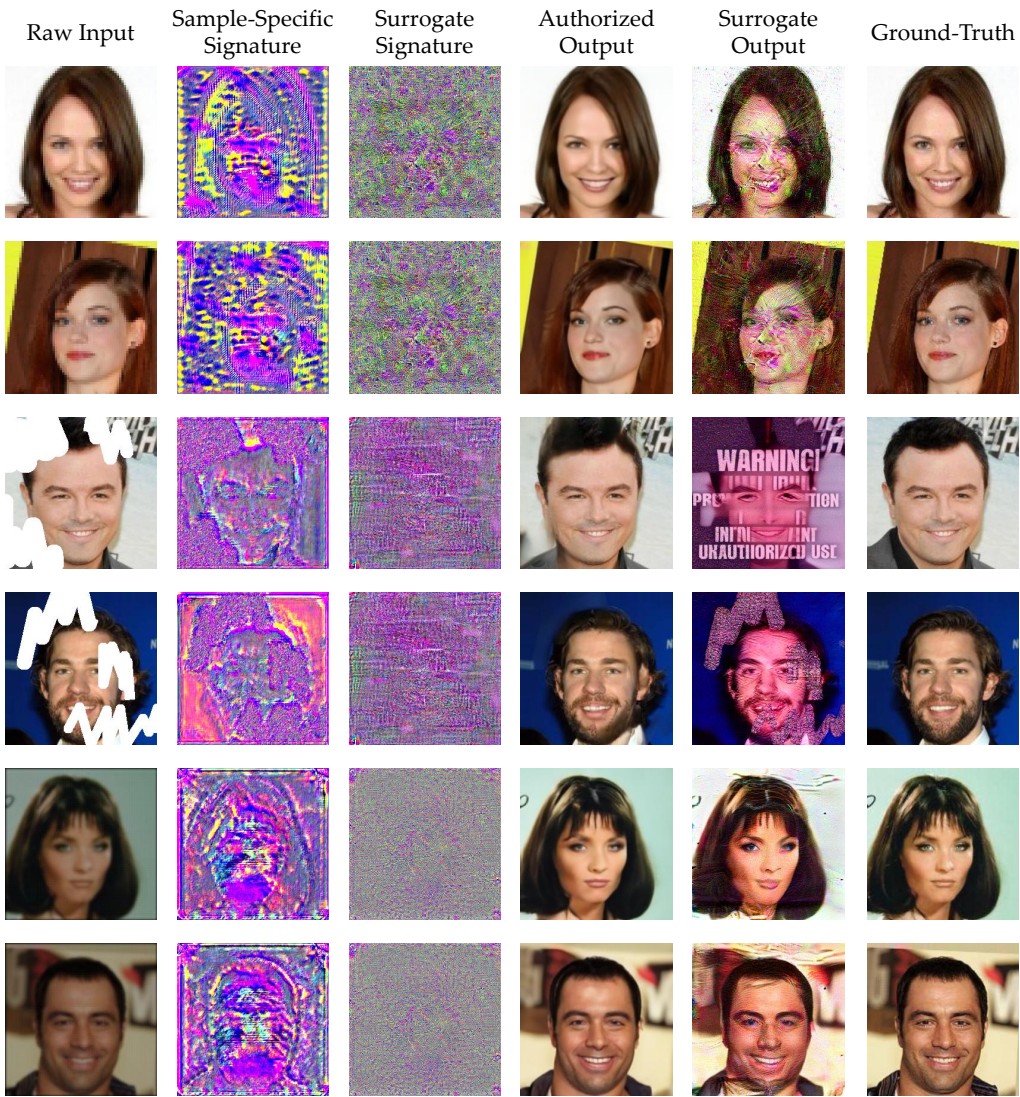

Raw Input | Sample-Specific Signature | Surrogate Signature | Authorized Output | Surrogate Output | Ground-Truth

*Figure 12.* **Visualization Results for Security Analysis: Sample-Specific Signature Setting.** We implement the sample-specific signature setting on the CelebA dataset with the I2SB bridge model. **Top 2 rows**: Super-Resolution task. **Middle 2 rows**: Inpainting task. **Bottom 2 rows**: Deblurring task. The results show that the adversary fails to recover a universal surrogate signature. The generation results for unauthorized inputs remain low-quality, indicating the sample-specific signature setting is more secure.

*Table 9.* **ImageNet: Protection Effectiveness (PE) vs Generation Quality (GQ).** For each task and bridge model, we report two adjacent rows: unprotected model (Unprotected) and *GoodDiffusion* protected model. The protection effectiveness is measured by **Abuse Rate** (AR), which indicates the fraction of unauthorized inputs incorrectly mapped to high-quality target images. As the unprotected model does not perform any protection, the AR is always 100%. The generation quality is evaluated by **FID**, **PSNR**, and **SSIM**. We also report **Error Rate** (ER), which measures the fraction of authorized generation incorrectly blocked to warning images.

| Task | Bridge Model | Setting | PE | GQ | | | |
| | | | Abuse Rate↓ | FID↓ | PSNR↑ | SSIM (E-02)↑ | Error Rate↓ |
|---|---|---|---|---|---|---|---|
| SR | DDBM-VP | Unprotected | 100 | 29.55 | 25.77 | 73.62 | – |
| | | *GoodDiffusion* | 0 | 36.04 | 25.67 | 70.61 | 0 |
| | DDBM-VE | Unprotected | 100 | 26.73 | 27.11 | 77.94 | – |
| | | *GoodDiffusion* | 0 | 34.50 | 24.05 | 75.34 | 0 |
| | I2SB | Unprotected | 100 | 28.22 | 27.85 | 78.89 | – |
| | | *GoodDiffusion* | 0 | 31.62 | 26.25 | 74.80 | 0.31 |
| | DBIM | Unprotected | 100 | 33.56 | 26.15 | 76.71 | – |
| | | *GoodDiffusion* | 0 | 43.41 | 25.40 | 73.65 | 0.31 |
| Inpaint | DDBM-VP | Unprotected | 100 | 22.28 | 24.60 | 81.97 | – |
| | | *GoodDiffusion* | 0 | 34.81 | 22.38 | 80.64 | 0 |
| | DDBM-VE | Unprotected | 100 | 27.53 | 23.61 | 81.33 | – |
| | | *GoodDiffusion* | 0 | 37.03 | 21.82 | 77.88 | 0 |
| | I2SB | Unprotected | 100 | 14.13 | 25.21 | 87.17 | – |
| | | *GoodDiffusion* | 0 | 15.23 | 24.44 | 85.87 | 0 |
| | DBIM | Unprotected | 100 | 22.91 | 22.18 | 83.86 | – |
| | | *GoodDiffusion* | 0 | 47.45 | 20.32 | 81.12 | 0 |
| Deblur | DDBM-VP | Unprotected | 100 | 4.82 | 35.98 | 94.89 | – |
| | | *GoodDiffusion* | 0 | 5.70 | 34.10 | 93.23 | 0 |
| | DDBM-VE | Unprotected | 100 | 8.53 | 34.61 | 94.11 | – |
| | | *GoodDiffusion* | 0 | 17.67 | 30.45 | 86.10 | 0 |
| | I2SB | Unprotected | 100 | 29.71 | 28.35 | 79.74 | – |
| | | *GoodDiffusion* | 0 | 42.47 | 26.02 | 73.42 | 0 |
| | DBIM | Unprotected | 100 | 3.31 | 37.69 | 96.74 | – |
| | | *GoodDiffusion* | 0 | 6.45 | 34.79 | 94.89 | 0 |

