# OpenReview forum: "GoodDiffusion: Proactive Copyright Protection for Diffusion Bridge Models via Learnable Sample-specific Signatures"
_ICML.cc/2026/Conference — ICML 2026 spotlight_

### Official Review · Reviewer_mzH6 · 2026-03-10

**Soundness:** 4
**Presentation:** 4
**Significance:** 4
**Originality:** 3
**Overall Recommendation:** 6
**Confidence:** 4

**Summary:**

The paper proposes a copyright protection method for diffusion bridge models. This method effectively denies unauthorized requests while only slightly degrading outputs for authorized requests. Heavily based on backdoor attack mechanisms, the method introduces a learnable signature network for sample-specific triggers, which can defend against white-box scenarios.

**Compliance With Llm Reviewing Policy:**

Affirmed.

**Final Justification:**

All my concerns have been resolved. Therefore, I maintain my recommendation.

**Key Questions For Authors:**

1. Can the thresholds for failed protection and occurred errors in all experiments be provided? Since AR values could be artificially tuned to obtain more favorable results, reporting these thresholds is critical for transparency. Furthermore, are these thresholds determined by some mechanisms or rules?
2. In Figure 4, the static surrogate does not perceptually resemble the static signature. Quantitatively, how far apart are these two signatures?

**Limitations:**

The most significant limitation of the proposed method lies in the applicable generative models. Although it is claimed that the proposed method is applicable to diffusion generative models, supporting evidence is absent. Therefore, the proposed method should be considered applicable only to diffusion bridge models.

**Strengths And Weaknesses:**

**Strength**
- An important but well-studied problem is revisited, and a solution from a completely distinct perspective is proposed. Although the solution is strongly motivated by backdoor attacks, the proposed method is further refined to address an identified vulnerability, thereby genuinely achieving its intended goal.
- The extensive experiments verify the effectiveness of the proposed method. The experimental design is comprehensive, demonstrating that the proposed method achieves the two protection goals.
- From Figure 1, readers can obtain a high-level understanding of the proposed method and its differences from watermarking methods, which is beneficial for readers.

**Weakness**
- In several places (e.g., the title and the first sentence of the abstract), the object of protection should be diffusion bridge models rather than *general* diffusion models, since the former are the primary focus of the study and the latter are not discussed.

---

> ### Author Rebuttal · Authors · 2026-03-31
>
> We sincerely express our gratitude for your acceptance and constructive comments. Below are our responses to your questions and identified weaknesses:
>
> > Q1: In several places (e.g., the title and the first sentence of the abstract), the object of protection should be diffusion bridge models rather than general diffusion models, since the former are the primary focus of the study and the latter are not discussed.
>
> **A1:** Thank you for your suggestion. We will revise the expressions to specify that our method is designed for diffusion bridge models. However, we would like to clarify that our method is not limited to diffusion bridge models and can be applied to general diffusion models as well. **We conduct some preliminary experiments on T2I diffusion models**. The core idea is to follow the backdoor attack methods in T2I models: a special text prompt is used as the signature, and the model solely generates high-quality images when the signature is present in the text prompt. We compute the FID between the generated images and the randomly selected samples from MS-COCO, and the CLIP Score between the generated images and the text prompts.
>
> The results show that GoodDiffusion can also provide effective copyright protection for T2I diffusion models. Kindly note that we include some T2I visualizations in the following link: https://anonymous.4open.science/r/ICML26-9548. For visualizations, **Fig. 1** and **Fig. 2** show the authorized T2I generation examples and **Fig. 3** and **Fig. 4** show the corresponding unauthorized examples.
>
> |Model|Settings|FID|CLIP Score|ER (%)|AR (%)|
> |-|:-:|:-:|:-:|:-:|:-:|
> |SD V1.4|Unprotected|50.99|31.22|-|-|
> |SD V1.4|Authorized|50.62|30.98|0.6|-|
> |SD V1.4|Unauthorized|284.26|17.21|-|3.3|
> |SD V1.5|Unprotected|50.00|31.24|-|-|
> |SD V1.5|Authorized|50.70|31.07|0.7|-|
> |SD V1.5|Unauthorized|282.63|17.83|-|4.5|
>
> > Q2: The paper should clarify how the thresholds for ER and AR are set.
>
> **A2:** We follow the metric of ASR in previous backdoor attack works [1, 2] to evaluate the AR and ER of our method. Specifically, the threshold for AR is set based on the mean MSE between the generated images and the target warning image. We use this value as a reference to select generated images that are far from the warning image. Then we conduct human evaluation to check whether the selected generated images bypass our protection. Similarly, the threshold for ER is set in a similar way, where we calculate the mean MSE between the images generated by the model and the original target images, and use this value as a reference to select generated images that are not close to the original target images, and verify these images through human evaluation. We will clarify this point in our revised manuscript.
>
> > Reference:
> >
> > [1] Zhai, S., Dong, Y., Shen, Q., Pu, S., Fang, Y., & Su, H. "Text-to-image diffusion models can be easily backdoored through multimodal data poisoning." ACM MM (2023).
> >
> > [2] Chen, W., Song, D., & Li, B. "Trojdiff: Trojan attacks on diffusion models with diverse targets." CVPR (2023).
>
> > Q3: In Figure 4, the static surrogate does not perceptually resemble the static signature. Quantitatively, how far apart are these two signatures?
>
> **A3:** We use the MSE to quantitatively evaluate the distance. In particular, we analyze the surrogate signature of the diffusion bridge model trained for the deblurring task. The table reports the MSE between the static surrogate and the static signature (MSE-S) and the MSE between random noise and the true static signature (MSE-R). The results show that the MSE-S is much smaller than the MSE-R. This suggests that the surrogate captures the essential features of the static signature. Thus, the static signature can be effectively approximated by a surrogate signature.
>
> |**Model**|**MSE-R**|**MSE-S**|
> |-|:-:|:-:|
> |DDBM-VP|1.01|0.30|
> |DDBM-VE|1.01|0.11|
> |I2SB|1.03|0.05|

---

> > ### Author Rebuttal · Reviewer_mzH6 · 2026-04-02
> >
> > My concerns have all been addressed. Therefore, I maintain my positive score and recommend that this paper be accepted at this venue.

---

> > > ### Author Response · Authors · 2026-04-02
> > >
> > > Thank you for your time and evaluation. We appreciate your encouraging comments and helpful suggestions.
> > >
> > > We will incorporate the clarifications from our rebuttal and refine the manuscript.

---

### Official Review · Reviewer_mjQ3 · 2026-03-10

**Soundness:** 3
**Presentation:** 4
**Significance:** 2
**Originality:** 3
**Overall Recommendation:** 4
**Confidence:** 3

**Summary:**

This paper tries to develop a proactive copyright protection against unauthorized use of diffusion generative models. Inspired by backdoor attacks, they preserve authorized queries and refuse unauthorized inputs by inducing different diffusion trajectories. After thorough analyses of naïve static-signature designs, they address potential safety problem by introduce a Learnable Signature Network to assign sample-specific signatures. Experiments validate the effectiveness of proposed GoodDiffusion.

**Compliance With Llm Reviewing Policy:**

Affirmed.

**Final Justification:**

I think this is a well-written paper with minimal flaws, and should be present in the ICML conference.

**Key Questions For Authors:**

1.	Is it possible to generalize such signatures control from I2I to T2I models?

2.	I am wondering whether it’s possible to generalized warning image from identical to dynamic. I believe this could be beneficial for model utility.

**Limitations:**

Authors should talk about potential limitations of their method, like adaptive attacks, computational overheads, image degradation, etc.

**Strengths And Weaknesses:**

Strengths:

1.	The proactive protection mechanism for diffusion models seems innovative.

2.	Authors build a clear threat model and try to discuss on real-world applications, which is welcome.

3.	Thorough analyses on naïve signature injection method increase the integrity of this paper, and make the idea of sample-specific signatures emerge naturally.

4.	Presentations and readability are good.

Weaknesses:

1.	One of my major concerns on this paper is the practical scenario. The authors said “the authorized user requests valid, sample-specific signatures from the model owner for signature injection and then runs the diffusion model”, however, I don’t think it is necessary to add a backdoor perturbation for preventing unauthorized generation. Model owner can just check the metadata of potential users for verification, and simply reject unauthorized users. Furthermore, when a model thief infiltrates the user’s sever, I don’t know why they cannot steal legitimate signatures. As you said, these signatures may be maintained by the model owner, but how the model owner can distinguish between an authorized user and an (infiltrated) authorized users because username/password/etc are all stolen by a malicious attacker?

2.	The proposed method can still be vulnerable to potential signature extraction. Although they are sample-specific, attackers can still use some adaptive attacks like introducing potential semantic extractor on N by modifying Eq. (5). The authors didn’t discuss about potential adaptive attacks. Moreover, as you assume that attackers can access to model architectures and parameters, why they can’t access to the Learnable Signature Network $g_\phi$? With the knowledge of $g_\phi$, I believe it is quite easy for attackers to recover the signatures.

3.	Some experimental results are still not that satisfactory. For instance, FID demonstrates some degradations for all models as shown in Table 1.

4.	Lacks details on training hyperparameters, compute resources, or code availability.


I would like to reconsider the rating if the authors could address my concerns well (Especially for W1 and W2).

---

> ### Author Rebuttal · Authors · 2026-03-31
>
> We sincerely express our gratitude for your acceptance and constructive comments. Below are our responses to your questions and identified weaknesses:
>
> > Q1: The necessity of adding backdoor perturbation for model-level protection and concerns about the security of signatures in the case of server infiltration.
>
> **A1:** Thank you for your question. We acknowledge that **software-level** protections (e.g., user verification) can be effective, but GoodDiffusion provides an additional layer of **model-level** protection, which is independent of the software-level defenses. Please refer to our response to **Reviewer 4YP2-Q5** for a detailed comparison between software-level and model-level defenses.
>
> As for the security of signatures, we assume that **the attacker can only access the diffusion model, but not the signature service.** You can refer to our response to **Reviewer KxT4-Q2** for a detailed discussion on the assumption.
>
> > Q2: Potential adaptive attacks for the sample-specific signatures.
>
> **A2:** We implement an adaptive attack to evaluate the robustness of GoodDiffusion against such attacks. Specifically, we randomly initialize a surrogate LSN $g_{\phi'}$ and freeze the parameters of a well-trained I2SB GoodDiffusion model $\epsilon_{\theta}$. **A straightforward attack is to modify Eq. (5)**:
>
> $$\mathcal{L}(\phi')=\mathbb{E}\|\epsilon_{\theta}(\hat{x}_t,t)-\epsilon^*(\hat{x}_t,t)\|$$
>
> The sample-specific signature is injected into the input of the model through the surrogate LSN $g_{\phi'}$.
>
> We train the surrogate LSN for 200 steps with a batch size of 256, and compare the performance of the model with the original LSN (O-LSN) and the surrogate LSN (S-LSN). The results show that the surrogate LSN fails to substitute the original LSN to bypass the protection, as the model can only generate the warning images.
>
> |**Model**|**FID**|**PSNR**|**SSIM**|**ER**|
> |-|:-:|:-:|:-:|:-:|
> |O-LSN|28.25|30.72|86.18|0.25|
> |S-LSN|376.91|6.04|5.42|100|
>
> We think the main reason for the results is that the GoodDiffusion model is trained to exhibit a threshold behavior: unless the transformed input is close enough to the authorized manifold, the model will generate warning images. **Such behavior is hard to learn without the supervision of the signature service.** We will include this experiment in our revised manuscript.
>
> > Q3: Some experimental results are still not that satisfactory.
>
> **A3:** We agree that the GoodDiffusion may cause some performance degradation compared to the original model, as discussed in Section 5.3. As we integrate two generation branches (the authorized branch and the warning branch) into a single model, there is a reasonable trade-off between copyright protection and generation quality. However, we believe that GoodDiffusion serves a more important purpose of copyright protection regardless of the performance degradation. In addition, it is common for the model-level protection methods to cause some performance degradation (e.g., watermarks).
>
> > Q4: Lacks details on training hyperparameters, compute resources, or code availability.
>
> **A4:** We apologize for the lack of details in the original submission. Please refer to our response to **Reviewer KxT4-Q1** for the hyperparameters and our response to **Reviewer 4YP2-Q2** for the compute resources. We will also provide code availability in our revised manuscript.
>
> > Q5: Is it possible to generalize such signature control from I2I to T2I models?
>
> **A5:** We conduct some preliminary experiments on T2I diffusion models to evaluate the generalization of GoodDiffusion. Please refer to our response to **Reviewer mzH6-Q1** for the details.
>
> > Q6: I am wondering whether it’s possible to generalize a warning image from identical to dynamic. I believe this could be beneficial for model utility.
>
> **A6:** Thank you for your suggestion. To implement dynamic warning images, we can prepare a set of warning images {$\bar x_1^1, \bar x_1^2, \ldots, \bar x_1^k$}. During the GoodDiffusion training, we can randomly select one warning image from the set for each training sample. In this way, the model learns to generate different warning images for unauthorized inputs, which may further enhance the protection effect.
>
> However, we believe that it is also a trade-off between copyright protection and the warning effect. While dynamic warning images may be more effective for copyright protection, such designs may bring some degradation effect on the generation quality. Overall, it is a promising research direction to explore in the future, and we will consider it as one of our future works.

---

> > ### Author Rebuttal · Reviewer_mjQ3 · 2026-03-31
> >
> > Thanks for your rebuttal. My major concerns have been solved. Therefore, I'd like to increase my rating to 4.
> >
> > The reason that why not a higher rating may raise from intrinsic drawbacks: The proposed method relies on training/finetuning on diffusion models with certain datasets, which may undermine their practical scenarios, and the qualilty degradation also compromises.
> >
> > Anyway, I still think this is a well-written paper with minimal flaws, and should be present in the ICML conference.

---

> > > ### Author Response · Authors · 2026-04-01
> > >
> > > Thank you sincerely for your timely review and comments.
> > >
> > > We truly appreciate your recognition of our work. Accordingly, we will incorporate the clarifications from our rebuttal into the revised manuscript. Moreover, we will further explore proactive defense mechanisms for generative models to resolve the existing drawbacks.
> > >
> > > Thank you again for your valuable feedback.

---

### Official Review · Reviewer_4YP2 · 2026-03-12

**Soundness:** 3
**Presentation:** 4
**Significance:** 3
**Originality:** 3
**Overall Recommendation:** 4
**Confidence:** 4

**Summary:**

This paper addresses the critical issue of proactive copyright protection for diffusion-generative models. The authors identify the core limitations of existing defense mechanisms and propose GoodDiffusion, a novel model-level proactive defense framework inspired by backdoor attack mechanisms. The paper demonstrates that naive static signatures are extremely vulnerable in white-box scenarios. Furthermore, to defend against this white-box threat, the authors introduce a learnable signature network (LSN). The LSN can generate unique, sample-specific signatures conditioned on each input image. The experimental method maintains high generation quality and achieves a near-zero abuse rate.

**Compliance With Llm Reviewing Policy:**

Affirmed.

**Final Justification:**

Thank you for the detailed rebuttal. My concerns have been resolved. The explanations regarding computational resource consumption and the assumptions made in the paper are all reasonable. Thus I raise my score.

**Key Questions For Authors:**

- Did the authors test the robustness of the model to fine-tuning attacks under warning branches?
- Core I2I tasks like image inpainting inherently involve irreversible degradation (information loss through masking), and the natural image lies on a nonlinear, non-Gaussian manifold. Does the proof hold in this case?
- Given that most business models utilize cloud APIs, traditional gateway authentication is sufficiently secure. What are the advantages and significance of the method presented in this paper compared to traditional approaches?

**Limitations:**

Yes.

**Strengths And Weaknesses:**

## Strengths

- The paper proposes a novel approach that combines backdoor attacks with copyright protection, achieving the effect of proactive data copyright protection.
- The experimental results of this paper are surprising; the good diffusion model achieves a high defense rate without affecting the performance of downstream tasks.
- The paper explicitly points out the vulnerability of static signatures and uses LSN to generate sample-specific signatures, which greatly increases the difficulty of white-box cracking.

## Weaknesses

- The paper doesn't comprehensively consider the infringer's capabilities. For example, it doesn't address whether infringers using clean and degraded sample pairs to fine-tune the model would reduce the effectiveness of the GoodDiffusion defense. This common scenario requires testing.

- The scenario described in the paper needs to consider the computational resources and time costs incurred by the model service provider when adding signatures to determine whether it meets the requirements of the actual application scenario. Currently, the paper lacks such experiments.

- Assumption A.3 of the paper, which assumes that images follow a Gaussian distribution, is unreasonable. Real-world images are highly complex, therefore this assumption is flawed.

---

> ### Author Rebuttal · Authors · 2026-03-31
>
> We sincerely express our gratitude for your acceptance and constructive comments. Additional tables and visualizations are included in the following link: https://anonymous.4open.science/r/ICML26-9548. Reference to **Tab. x** in the following responses corresponds to the table in the above link. Below are our responses to your questions and identified weaknesses:
>
> > Q1: If infringers using clean and degraded sample pairs to fine-tune the model would reduce the effectiveness of the GoodDiffusion defense?
>
> **A1:** We fine-tune a well-trained GoodDiffusion model for 200 steps with a batch size of 256 image pairs. For comparison, we also train a randomly initialized model with the same settings. The results are shown in **Tab. 1**. As shown in the table, the model trained from scratch performs much better than the fine-tuned GoodDiffusion model.
>
> Overall, we assume that **the infringers may not have the resources to fine-tune the diffusion model.** Even if they do, the performance of the fine-tuned model is still worse than a model trained from scratch. Thus, **the infringers do not have to steal the model but train a new one from scratch.**
>
> > Q2: The computational resources and time costs?
>
> **A2:** We report the maximum GPU memory usage and inference time of GoodDiffusion in **Tab. 2**. The results are obtained by running the model on an RTX 3090 GPU with a batch size of 16 and a resolution of 256x256 for 1 inference step.
>
> The results show that the LSN does not bring significant computational overhead. In addition, as image generation requires a number of steps for the diffusion model, but only one inference for the LSN, the additional computation and time cost of LSN are negligible.
>
> > Q3: Assuming images follow a Gaussian distribution is unreasonable.
>
> **A3:** We agree that real-world images are complex and each image may not strictly follow a Gaussian distribution in the pixel space. However, our Assumption A.3 is a common simplification under appropriate representations. According to the **central limit theorem (CLT)**, the i.i.d. images can be approximated as Gaussian distributions in most cases, which is aligned with classical findings in the image modeling literature [1, 2]. Moreover, the Gaussian assumption is only used for the theoretical analysis to derive the score function, while our main conclusions are empirical and do not rely critically on exact Gaussianity. We will clarify this point in our revised manuscript.
>
> > Reference:
> >
> > [1] Ruderman, D. L. "The statistics of natural images." Network: computation in neural systems (1994).
> >
> > [2] Simoncelli, E. P., & Olshausen, B. A. "Natural image statistics and neural representation." Annual review of neuroscience (2001).
>
> > Q4: Core I2I tasks involve irreversible degradation, and the natural image lies on a nonlinear, non-Gaussian manifold. Does the proof hold in this case?
>
> **A4:** We agree the current theoretical analysis is written in a simplified setting and may not directly apply to real-world scenarios. We extend Theorem 4.1 to a more general setting. Consider a general corruption operator $f$:
> $$x_0=f(x_1)$$
> where $f$ can be nonlinear and irreversible and $x_0,x_1$ may not follow a Gaussian distribution. According to Lemma A.1, we can derive the conditional mean of $x_t$ as:
> $$\mu_t(x_1)=a_tx_1+b_tf(x_1)$$
> Thus, using the first-order Taylor expansion around $x_1$, we have:
> $$\mu_t(x_1+k)-\mu_t(x_1+N)\approx(a_tI+b_t\frac{\partial f}{\partial x_1})(k-N)$$
> Therefore, matching the two trajectories drives the optimizer to solve the equation:
> $$(a_tI+b_t\frac{\partial f}{\partial x_1})(k-N)=0$$
> As long as the matrix $a_tI+b_t\frac{\partial f}{\partial x_1}$ is non-singular, the only solution to the above equation is $k=N$. We will include this extended analysis in our revised manuscript.
>
> > Q5: What are the advantages and significance of the method?
>
> **A5:** The main advantages of our method compared to traditional software-based defenses are:
>
> 1. **Model-level protection**: We acknowledge that the gateway authentication is a powerful **software-level** defense. However, the gateway authentication does not protect the model once the model is leaked. In this paper, we do not intend to replace the gateway authentication but rather to complement it: our method provides an additional layer of **model-level** protection. As the protection is embedded in the generative process, even if the model is stolen, the unauthorized users will still be unable to generate high-quality images without valid signatures. Overall, our method can be integrated with existing gateway authentication to provide a more comprehensive defense strategy.
> 2. **Proactive defense at generation time**: Most existing model-level defenses, such as watermarking, are **passive** which only allow for post-hoc detection of unauthorized use. In contrast, our method is a **proactive** defense that directly drives unauthorized queries to a warning image, refusing unauthorized generation.

---

> > ### Author Rebuttal · Reviewer_4YP2 · 2026-04-03
> >
> > Thank you for the detailed rebuttal. My concerns have been resolved. The explanations regarding computational resource consumption and the assumptions made in the paper are reasonable. Thus I raise my score to 4.

---

> > > ### Author Response · Authors · 2026-04-03
> > >
> > > Thank you very much for your time and thoughtful evaluation. We sincerely appreciate your positive feedback.
> > >
> > > We are glad that our response helped clarify the concerns. We will incorporate the rebuttal into the revised manuscript.

---

### Official Review · Reviewer_KxT4 · 2026-03-13

**Soundness:** 2
**Presentation:** 3
**Significance:** 2
**Originality:** 3
**Overall Recommendation:** 4
**Confidence:** 3

**Summary:**

This paper proposes a proactive copyright protection mechanism GoodDiffusion that cuts off unauthorized use of diffusion generative models, which internalizes authorization into the generative process rather than relying on post-hoc attribution like watermarking or fingerprinting. The protection goals can be summarized as: 1) generate high-quality images for inputs carrying valid signatures, 2) refuse to generate for unauthorized inputs. To achieve the goal, a naive baseline based on the backdoor attack mechanism is proposed to produce high-quality outputs only when the input carries a predefined signature. However, the fixed signature for all samples are proven to be fragile in practice since the signature could be recovered via gradient-based optimization. To strengthen security, the Learnable Signature Network (LSN) is proposed to learn a specific signature conditioned on each input.

**Compliance With Llm Reviewing Policy:**

Affirmed.

**Final Justification:**

I keep my score

**Key Questions For Authors:**

The same as the previous Weaknesses section.

**Limitations:**

yes

**Strengths And Weaknesses:**

### Strengths
1. This paper is an early trial for proactive copyright protection, shifting post-hoc output-level protection to model-level prevention. The pinciple of ​​refusing generation at the earliest possible stage is meaningful.
2. The intuition of a fixed signature can be interpreted as a fixed vector in the pixel space, leading to a parallel shift of the entire image manifold while the sample-specific signatures produce a nonlinearly deformed image manifold is easy to understand.
3. Extensive experiments of three representative Image-to-Image tasks (i.e., super-resolution, inpainting, and deblurring) on CelebA and ImageNet datasets with various diffusion bridge models (i.e., DDBM-VP, DDBM-VE, I2SB, DBIM) to validate the effectiveness of GoodDiffusion.

### Weaknesses
1. In Eq. 6, GoodDiffusion utilizes a hyperparameter to controls the strength of the signature injection, however, the effect of this hyperparameter is missed.
2. Is the assumption "the thief cannot access the legitimate signature service" too strong or not? There is no supporting literature.
3. Even without access to the original LSN, can an attacker use surrogate learning to simulate an LSN? For example, Prior's work on model stealing demonstrates that attackers can approximate protected services through repeated queries (i.e., https://arxiv.org/abs/1812.02766). It would be even better if there is relevant experiments.

---

> ### Author Rebuttal · Authors · 2026-03-31
>
> We sincerely express our gratitude for your acceptance and constructive comments. Below are our responses to your questions and identified weaknesses:
>
> > Q1: In Eq. 6, GoodDiffusion utilizes a hyperparameter to control the strength of the signature injection; however, the effect of this hyperparameter is missed.
>
> **A1:** We train the I2SB model for the Super-Resolution task on the Celeba dataset with different values of $\gamma$ to evaluate the effect of the hyperparameter.
> ||**AR (%)**|**FID**|**PSNR**|**SSIM**|**ER (%)**|
> |-|:-:|:-:|:-:|:-:|:-:|
> |Unprotected|100|22.05|32.48|89.93|-|
> |$\boldsymbol{\gamma}=0.99$|15.88|32.93|28.38|79.48|8.44|
> |$\boldsymbol{\gamma}=0.9$|0|**28.25**|**30.72**|**86.18**|0.25|
> |$\boldsymbol{\gamma}=0.7$|0|33.65|27.18|79.93|0|
> |$\boldsymbol{\gamma}=0.5$|0|37.24|25.66|75.88|0|
> |$\boldsymbol{\gamma}=0.3$|0|42.91|23.41|70.52|0|
> |$\boldsymbol{\gamma}=0.1$|0|57.38|19.61|61.83|0|
>
> The results show that as $\gamma$ decreases, the quality of the generated images degrades, which aligns with our intuition that a smaller $\gamma$ leads to a stronger signature injection, thus ruining the information in the input images. However, if $\gamma$ is too large (e.g., $\gamma=0.99$), the model is hard to distinguish the authorized and unauthorized inputs as the signature injection is too weak, thus leading to a high AR/ER and affecting the generation quality. Overall, to achieve a good generation quality, we choose $\gamma=0.9$ in our experiments.
>
> > Q2: Is the assumption "the thief cannot access the legitimate signature service" too strong or not? There is no supporting literature.
>
> **A2:** Thank you for your question. In this paper, we do not intend to build an omnipotent defense that can protect the copyright in all circumstances. Instead, we only focus on **a specific threat model**: the adversary may obtain the diffusion model from an authorized user **without simultaneously** accessing the signature service. We consider this assumption to be reasonable as it is aligned with the "separation of duties" principle in security [1]. That is, the signature service is maintained separately by the model owner, while the model is deployed on the authorized user’s server. Even if an attacker can access the model, they cannot access the signature service without also compromising the owner’s infrastructure. This separation is a common security practice to mitigate potential threats. In practice, modern commercial Key Management Service (KMS) implements such separation by design, where the clients obtain authorization from a centralized license server rather than carrying the signature service locally [2,3,4]. Overall, we believe that this assumption is reasonable and reflects a common real-world scenario. We will clarify this assumption in our revised manuscript with supporting references.
>
> > Reference:
> >
> > [1] Groll, S., Fuchs, L., & Pernul, G. "Separation of duty in information security." ACM Computing Surveys (2025).
> >
> > [2] Google Cloud. "Cloud Key Management Service (Cloud KMS)." <https://docs.cloud.google.com/kms/docs/separation-of-duties>.
> >
> > [3] Google Cloud. "Cloud Security." <https://docs.cloud.google.com/docs/security/key-management-deep-dive>.
> >
> > [4] Barker, E. B., Barker, W. C., Burr, W. E., Polk, W. T., & Smid, M. E. "Recommendation for key management, part 1: General (revised)." (2007).
>
> > Q3: Even without access to the original LSN, can an attacker use surrogate learning to simulate an LSN?
>
> **A3:** We acknowledge that the diffusion model may be vulnerable to some model extraction attacks. To evaluate the robustness of GoodDiffusion against such attacks, we conduct an additional experiment to simulate the surrogate learning attack in the CelebA Super-Resolution task. As we clarify in **A2**, we assume that **the attacker cannot access the signature service**. Thus, we can only repeatedly query a well-trained I2SB model $\epsilon^*$ to train a diffusion new model $\epsilon'$:
>
> $$\mathbb{E}_{x \sim \mathcal{D}}\|\epsilon'(x_t,t)-\epsilon^*(x_t,t)\|.$$
>
> In this way, the new model $\epsilon'$ may approximate the generation behavior of $\epsilon^*$ by repeated queries. However, as the distillation lacks the supervision from the signature service, **the surrogate learning attack cannot learn the exact signature injection behavior**, thus $\epsilon'$ fails to bypass the protection of GoodDiffusion. The results below show that the surrogate model $\epsilon'$ can only generate the warning images, but fails to generate high-quality images for authorized generation. In contrast, the original model can generate high-quality images when the authorized signature is present.
>
> |**Model**|**FID**|**PSNR**|**SSIM**|
> |-|:-:|:-:|:-:|
> |$\epsilon^*$|28.25|30.72|86.18|
> |$\epsilon'$|372.31|5.89|5.16|
>
> In addition, we conduct an adaptive attack by training a surrogate LSN $g_{\phi'}$ to approximate the original LSN $g_{\phi}$ in our response to **Reviewer mjQ3-Q2**.

---

> > ### Author Rebuttal · Reviewer_KxT4 · 2026-04-01
> >
> > Thanks for your rebuttal. My concerns have been solved. Therefore, I will keep my positive score.

---

> > > ### Author Response · Authors · 2026-04-02
> > >
> > > Thank you very much for your thoughtful review and constructive comments. We sincerely appreciate your positive feedback and recognition of our work. We will incorporate the improvements discussed in the rebuttal into the revised manuscript.

---

### Decision · Program_Chairs · 2026-04-30

**Decision:**

Accept (spotlight)

**Comment:**

In view of the observation that post-hoc methodologies like watermarking and fingerprinting offer only indirect and limited preventive effect, the authors in this paper propose GoodDiffusion, which is inspired by backdoor mechanisms, to enforce model-level use-time control by internalizing authorization into the generative process through a selectively permissive, otherwise closed behavior. To strengthen security, the authors introduce a Learnable Signature Network (LSN) to learn sample-specific signatures conditioned on each input.
After rebuttal, all reviewers consistently recognize the contribution of this paper, including (1) Proactive copyright protection that shifts post-hoc output-level protection to model-level prevention, (2) Novel approach that combines backdoor attacks with copyright protection, (3) Experimental results are promising, and (4) An important but well-studied problem is revisited, and a solution from a completely distinct perspective is proposed, and give positive scores.
BTW, all the concerns raised the reviewers are fully resolved.
Based on the above, the AC recommends this paper be accepted.